# *Antrodia cinnamomea* Formula Suppresses Prostate Cancer Progression via Immune Modulation and PD-1/PD-L1 Pathway Inhibition

**DOI:** 10.3390/ijms26062684

**Published:** 2025-03-17

**Authors:** Ming-Yen Tsai, Chung-Kuang Lu, Li-Hsin Shu, Hung-Te Liu, Yu-Huei Wu, Yu-Shih Lin, Yao-Hsu Yang, Wei-Tai Shih, I-Yun Lee, Yu-Heng Wu, Ching-Yuan Wu

**Affiliations:** 1Department of Chinese Medicine, Kaohsiung Chang Gung Memorial Hospital and Chang Gung University College of Medicine, Kaohsiung 83301, Taiwan; missuriae@cgmh.org.tw; 2Department of Chinese Medicine, Chiayi Chang Gung Memorial Hospital, Chiayi 613016, Taiwan; cmusiclck@gmail.com (C.-K.L.); pipi19880706@gmail.com (L.-H.S.); mfw914@gmail.com (H.-T.L.); irenewu7715@gmail.com (Y.-H.W.); r95841012@cgmh.org.tw (Y.-H.Y.); ati8955@cgmh.org.tw (W.-T.S.); ynalee@cgmh.org.tw (I.-Y.L.); 3Department of Pharmacy, Chiayi Chang Gung Memorial Hospital, Chiayi 613016, Taiwan; yohimba@cgmh.org.tw; 4School of Chinese Medicine, College of Medicine, Chang Gung University, Taoyuan 33302, Taiwan; 5Health Information and Epidemiology Laboratory, Chang Gung Memorial Hospital, Chiayi 613016, Taiwan; 6Institute of Communications Engineering, The National Yang Ming Chiao Tung University, Hsinchu City 300093, Taiwan; henrywu802@gmail.com; 7Center for Drug Research and Development, College of Human Ecology, Chang Gung University of Science and Technology, Taoyuan 33303, Taiwan

**Keywords:** XIANZHIFANG formula, prostate cancer, T-cells, PD-L1, PD-1, macrophage

## Abstract

Prostate cancer remains a significant global health challenge, necessitating the development of novel therapeutic approaches. This study investigated the therapeutic potential of the *Antrodia cinnamomea* formula (XIANZHIFANG formula, XZF), comprising *Antrodia cinnamomea*, *Sanghuangporus sanghuang*, *Ganoderma lucidum*, *Ganoderma sinense*, and *Inonotus obliquus*, in prostate cancer treatment. HPLC analysis confirmed the presence of key triterpenoids, including Antcin A, B, C, K, and Zhankuic acid B, C, and 4,7-dimethoxy-5-methyl-1,3-benzodioxole. Cytotoxicity assays demonstrated that XZF (50–200 μg/mL) exhibited selective activity, maintaining viability in non-cancerous 293T-cells while enhancing the viability of activated CD8+ and CD4+ T-cells in a dose-dependent manner. XZF significantly reduced PD-1 expression in CD8+ T-cells but not in CD4+ T-cells and inhibited the PD-L1/PD-1 interaction, achieving 93% inhibition at 200 μg/mL. Furthermore, when combined with atezolizumab (1 μg/mL), XZF demonstrated complete blockade of PD-L1/PD-1 interaction. In prostate cancer cells, XZF exhibited differential antiproliferative effects. In PC-3 cells, XZF significantly reduced viability across a concentration range of 25–200 μg/mL, whereas DU145 cells showed only partial inhibition at higher concentrations (100–200 μg/mL). LNCaP cells exhibited a dose-dependent reduction in viability, mirroring the response pattern of PC-3 cells. Conditioned medium from XZF-treated macrophages, particularly human THP-1 cells, significantly suppressed the viability and migration of prostate cancer cells in a dose-dependent manner. Notably, the conditioned medium from XZF-treated THP-1 cells exhibited a stronger inhibitory effect on prostate cancer cell viability and migration compared to murine RAW 264.7 macrophages. These findings indicate that XZF exerts its therapeutic potential through multiple mechanisms, including direct antiproliferative effects on cancer cells, enhancement of T-cell responses, modulation of immune checkpoint pathways, and macrophage-mediated suppression of prostate cancer cell survival and migration. The pronounced effects observed in human macrophage models suggest a promising avenue for further investigation in clinical settings, particularly in combination with existing immunotherapies.

## 1. Introduction

Globally, prostate cancer ranks as the second most frequently diagnosed malignancy [1]. This disease poses distinct obstacles in the realm of immunotherapy. Conventional immunotherapeutic methods, including the use of immune checkpoint inhibitors, have demonstrated only modest efficacy against prostate cancer. This limited effectiveness is primarily due to the cancer’s immunosuppressive tumor milieu and its relatively low mutational load, which diminishes the cancer’s ability to provoke an immune response [2]. The inherent resistance of prostate cancer to established immunotherapies, particularly immune checkpoint inhibitors, is closely associated with its specific tumor microenvironment [3]. This environment is predominantly marked by the presence of a high concentration of immunosuppressive cells and a scarcity of neoantigens, a consequence of the cancer’s low mutational frequency [4]. Often categorized as a “cold” tumor, prostate cancer is characterized by a subdued immune response, a robust immunosuppressive tumor environment, and insufficient recruitment of cytotoxic T-cells, thereby reducing the efficacy of conventional immunotherapies [5,6,7,8,9]. Nonetheless, the recent progress in decoding the immunological intricacies of prostate cancer has paved the way for innovative therapeutic strategies.

Contemporary research has illuminated the dualistic nature of macrophages in the advancement of prostate cancer. Macrophages demonstrate a dichotomy in function, serving both as facilitators and inhibitors of tumor growth, thereby underscoring their complex involvement in this malignancy [4]. Additionally, interactions between cancer-associated fibroblasts and macrophages have been identified as critical in the progression of prostate cancer [10]. The pro-tumorigenic activities of macrophages are postulated to be interconnected with their role in diminishing immune response during the development of prostate cancer [11,12]. This multifarious involvement of macrophages in prostate cancer exemplifies the wider intricacies faced in cancer immunotherapy, emphasizing the importance of deciphering and modulating the tumor immune microenvironment for the enhancement of therapeutic approaches.

In the realm of oncological research, the significance of immune checkpoint proteins, specifically Programmed Death-Ligand 1 (PD-L1) and Programmed Cell Death Protein 1 (PD-1), in the context of prostate malignancies has garnered considerable attention. These proteins serve as crucial mechanisms by which neoplastic cells circumvent immune system detection and destruction, a process often referred to as immune evasion [13,14]. The presence and upregulation of PD-L1 and PD-1 in prostate cancer correlate strongly with enhanced tumor malignancy, resistance to established therapeutic regimens, and a generally unfavorable patient prognosis [15,16,17]. The therapeutic landscape reveals that interventions targeting the PD-1/PD-L1 axis may represent a novel and effective strategy for combating this disease. Agents that inhibit PD-1 and PD-L1 have demonstrated encouraging outcomes in clinical trials, particularly in terms of patient survival rates and disease progression metrics [4]. Of notable interest is the synergistic potential observed when combining anti-PD-L1 agents with therapies aimed at mitigating myeloid-derived suppressor cell activity or inhibiting poly ADP-ribose polymerase, especially in cases of castration-resistant prostate cancer [18,19]. Despite these advances, the variability in patient responses underscores the necessity for continued research to unravel the complex molecular mechanisms at play and to identify predictive biomarkers that can aid in patient stratification and therapy customization. Additionally, there is a growing consensus on the exploration of combination therapeutic strategies that integrate PD-1/PD-L1 blockade with other modalities, aiming to amplify treatment efficacy and counteract resistance mechanisms [2,4]. A comprehensive understanding of the roles and interactions of PD-1 and PD-L1 within the tumor microenvironment of prostate cancer is imperative for the advancement of effective immunotherapeutic approaches and the enhancement of patient survival outcomes.

*Antrodia cinnamomea* formula (XIANZHIFANG formula, XZF), a popular health-promoting herbal formula in Taiwan, exemplifies an immunomodulatory dietary component rooted in traditional medicinal practices. This formulation is distinguished by its composition of a quintet of esteemed fungal species, each contributing unique bioactive properties. The assemblage includes *Antrodia cinnamomea* (AC), *Sanghuangporus sanghuang*, *Ganoderma lucidum*, *Ganoderma sinense*, and *Inonotus obliquus*. These fungi are recognized not only for their nutritional value but also for their potential therapeutic implications, particularly in modulating immune system responses. The synergy of these five fungal entities in the XZF positions it as a subject of interest in the field of nutraceuticals and integrative medicine, especially considering the growing emphasis on natural and holistic approaches to health enhancement and disease prevention.

The study aimed to comprehensively evaluate the potential effects of XZF through multiple mechanisms: its effects on CD4+ and CD8+ T-cell function, its impact on macrophage-mediated antitumor responses, and its ability to modulate the PD-L1/PD-1 pathway in both prostate cancer and T-cells. Of particular interest was the formula’s potential synergistic effects with established immunotherapy agents such as atezolizumab. This multifaceted investigation sought to establish a foundation for understanding XZF’s potential role in prostate cancer immunotherapy and its underlying mechanisms of action.

## 2. Results

### 2.1. Identification of Reference Compounds in XZF by HPLC Analysis

Because the primary components of XZF was *Antrodia cinnamomea*, the purified extract of *Antrodia cinnamomea* (AC) fruiting bodies with triterpenoids, including Antcin A, B, C, K, and Zhankuic acid B, C, and 4,7-dimethoxy-5-methyl-1,3-benzodioxole were used as the reference standard. Reverse phase High-performance liquid chromatography (HPLC) analysis of the approved XZF fingerprint chromatography (Figure 1A) confirmed the presence of indicated compounds (Figure 1B–I).

### 2.2. Effect of XZF on Cell Variability in the Non-Cancerous Cell Lines

To evaluate XZF’s selective cytotoxicity against cancer versus normal cells, human embryonic kidney 293T cells were selected as a non-cancerous control model. This widely used cell line exhibits stable growth characteristics and consistent behavior in culture systems, making it an ideal choice for assessing the general cellular toxicity of bioactive compounds. The cytotoxic effects were systematically evaluated using the 2,3-bis-(2-methoxy-4-nitro-5-sulfophenyl)-2H-tetrazolium-5-carboxanilide (XTT) assay, which provides a quantitative measure of cell viability through the assessment of metabolic activity. Treatment protocols were established to examine the concentration-dependent effects of XZF on cellular viability. The experimental design incorporated three distinct concentrations (50, 100, and 200 μg/mL) to establish a comprehensive dose-response relationship. Cell cultures were monitored over a 48-h period to capture both immediate and delayed effects of XZF exposure. The experimental results, as illustrated in Figure 2, revealed remarkable stability in cell viability across all tested concentrations. Statistical analysis of the XTT assay data demonstrated that exposure to XZF, even at the highest concentration of 200 μg/mL, did not induce significant alterations in cellular metabolic activity compared to untreated controls. The maintenance of cellular viability was consistent throughout the 48-h observation period, suggesting that XZF exhibits minimal cytotoxicity towards non-cancerous cells within the tested concentration range.

### 2.3. Effects of XZF on Cell Viability of T-Cells

Figure 2 delineates that the administration of XZF at concentrations of 50, 100, and 200 μg/mL did not exert a deleterious effect on the viability of 293T cells, thus highlighting the innocuous nature of XZF at these dosages. Within the composition of XZF, AC is identified as a key component known for its pronounced immunomodulatory properties, especially in the modulation of T-cell activity [20]. The presence of high-molecular-weight polysaccharides in AC is instrumental in enhancing dendritic cell functionality and in the activation of antigen-specific T-cells, thereby facilitating Th1 differentiation [21]. These attributes of AC are congruent with its prospective utilization in the domain of immunotherapy. Subsequent experiments were performed to evaluate the effects of XZF on CD8+ and CD4+ T-cell populations. Data indicated that exposure to XZF at concentrations of 50, 100, and 200 μg/mL resulted in a dose-dependent enhancement of viability in activated CD8+ T-cells within a 24-h timeframe, as depicted in Figure 3A. Correspondingly, activated CD4+ T-cells exhibited a dose-dependent increase in cell viability following a 24-h incubation with XZF at analogous concentrations, illustrated in Figure 3B. In conclusion, the findings of this research suggest that XZF, at concentrations of 50, 100, and 200 μg/mL, positively influences cell viability in both activated CD8+ and CD4+ T-cell subsets, indicating its supportive role in T-cell-mediated immune mechanisms.

### 2.4. Effects of XZF on PD-L1/PD-1 Pathway

The enhancement of PD-L1 and PD-1 expression on neoplastic and immunosuppressive cellular entities, inclusive of CD4+ and CD8+ T-cells, has been implicated in the diminution of CD8+ cytotoxic T-cell functionality. This overexpression, potentially indicative of an efficacious anti-neoplastic immune response, ultimately precipitates the functional exhaustion of tumor-specific CD8+ T-cells, as indicated by their elevated PD-1 expression, references [22,23]. To assess the influence of XZF on PD-1 expression levels, CD8+ and CD4+ T-cells were procured from murine splenic specimens. Post-isolation, these cells were subjected to a 24-h incubation with XZF at concentrations of 50, 100, and 200 μg/mL. Analysis thereafter revealed a marked diminution in the population of PD-1+ CD8+ T-cells following XZF treatment across all concentrations when contrasted with the control group, as depicted in Figure 4A. In contrast, PD-1+ CD4+ T-cell counts did not demonstrate significant variation post-treatment with analogous XZF concentrations, signifying a selective effect on CD8+ T-cells, as shown in Figure 4B.

Interferon-gamma (IFN-γ) is crucial in orchestrating both innate and adaptive immune responses, predominantly secreted by macrophages, T-cells, and natural killer cells [24]. While central to antitumor immune mechanisms, IFN-γ can paradoxically promote immune evasion, notably through the upregulation of PD-L1 [25,26]. In this research, the effect of XZF on the PD-L1/PD-1 pathway in T-cells and prostate cancer was investigated. Experimental outcomes demonstrated that XZF at concentrations of 100 and 200 μg/mL did not induce PD-L1 expression in PC-3 cells (Figure 5A,B) and DU145 cells (Figure 5C,D). Moreover, a 200 μg/mL concentration of XZF exhibited an inhibitory impact on PD-L1 expression, which was induced by 10 ng/mL of IFN-γ in PC-3 cells (Figure 5A,B) and DU145 cells (Figure 5C,D). Further exploration was directed towards examining the direct effect of XZF on PD-L1 and PD-1 binding, employing a PD-L1/PD-1 homogeneous assay. The results of this assay indicated a significant 93% inhibition of PD-L1/PD-1 interaction following the administration of 200 μg/mL of XZF. For comparison, atezolizumab at a concentration of 1 μg/mL achieved an inhibition rate of 98%. Notably, a combined regimen of 1 μg/mL atezolizumab and 200 μg/mL XZF led to a complete blockade of PD-L1 and PD-1 interaction, as demonstrated in Figure 5E. These findings suggest that XZF, particularly at a concentration of 200 μg/mL, substantially impedes the expression and interaction of PD-L1 and PD-1.

### 2.5. Effects of XZF on Cell Viability of Prostate Cancer Cells

Previous investigations assessed the effects of XZF at concentrations of 100 and 200 μg/mL, demonstrating that these concentrations did not induce PD-L1 expression in either PC-3 or DU145 prostate cancer cell lines, as illustrated in Figure 5A,B. Notably, at 200 μg/mL, XZF effectively attenuated PD-L1 expression that was otherwise upregulated by IFN-γ (10 ng/mL). This inhibitory effect was consistently observed in both PC-3 and DU145 cells, highlighting its potential therapeutic relevance. To further elucidate XZF’s impact, we conducted XTT assays to evaluate its effects on cell viability in PC-3, DU145, and LNCaP cells, the latter being an androgen-sensitive prostate cancer cell line. The results revealed distinct responses among these cell types. In PC-3 cells, XZF exhibited significant inhibitory effects across a concentration range of 25–200 μg/mL over 24–48 h (Figure 6A), indicating a pronounced antiproliferative activity. In contrast, DU145 cells displayed a more restricted response, with only partial inhibition of cell viability observed at higher concentrations (100–200 μg/mL) and becoming evident only after 48 h of exposure (Figure 6B). Moreover, LNCaP cells exhibited a dose-dependent reduction in viability within the 50–200 μg/mL range over the same time frame, showing a response pattern similar to that observed in PC-3 cells (Figure 6C). These findings suggest that XZF’s antiproliferative efficacy varies among prostate cancer cell lines, potentially due to intrinsic cellular differences that influence its mechanism of action.

### 2.6. The Effects of XZF Treatment on Prostate Cancer Cell Viability in Macrophage-Conditioned Medium

Macrophages have been shown to play crucial roles in prostate cancer progression, including their involvement in signaling pathways, therapy resistance, and modulation of immune checkpoint blockade therapies such as PD-1/PD-L1 targeted strategies [22,27,28,29]. Previously, we demonstrated that XZF treatment at 50, 100, and 200 μg/mL enhanced cell viability in specific lymphocyte populations, particularly CD8+ and CD4+ T-cells (Figure 3). To further investigate the effects of XZF on macrophage-mediated responses, we employed two macrophage cell lines—RAW 264.7 (Mus musculus) and THP-1 (Human)—and assessed cell viability using XTT assay. Treatment of RAW 264.7 cells with XZF (50, 100, and 200 μg/mL) for 24 h showed no significant impact on cell viability (Figure 7A). We next examined the effects of conditioned medium from XZF-treated RAW 264.7 cells on PC-3 prostate cancer cells. The conditioned medium was collected after a 24-h incubation of RAW 264.7 cells with or without XZF treatment. Our results revealed that conditioned medium from RAW 264.7 cells treated with higher concentrations of XZF (100 and 200 μg/mL) partially inhibited PC-3 cell viability at both 24 and 48 h (Figure 7B). Similarly, THP-1 cells exposed to XZF (50, 100, and 200 μg/mL) maintained their viability over 24–48 h (Figure 8A). However, the conditioned medium derived from XZF-treated THP-1 cells exhibited a more pronounced inhibitory effect on prostate cancer cell viability. Following a 24-h incubation period, the conditioned medium obtained from THP-1 cells treated with 100 or 200 μg/mL XZF significantly reduced the viability of PC-3 (Figure 8B) and DU-145 (Figure 8C) cells at both 24 and 48 h post-treatment. Additionally, the conditioned medium from THP-1 cells treated with 50, 100, or 200 μg/mL XZF led to a significant reduction in LNCaP cell viability at 24 h post-treatment (Figure 8D). These findings suggest that XZF may modulate macrophage-derived secretory factors, thereby indirectly influencing prostate cancer cell survival. Notably, the observed effects indicate a potentially greater impact in human macrophage models compared to murine counterparts.

### 2.7. Effects of XZF on Migration of Prostate Cancer Cells with Macrophage-Conditioned Medium

Building upon our previous findings that XZF modulates macrophage-secreted factors to influence prostate cancer cell survival, with notably stronger effects observed in human versus murine macrophage models, we proceeded to investigate XZF’s impact on macrophage-mediated prostate cancer cell migration. This investigation was particularly significant given the critical role of cell migration in cancer progression and metastasis. We employed a well-established transwell migration assay system to evaluate the effects of the macrophage-conditioned medium on prostate cancer cell migration. THP-1 cells were initially treated with varying concentrations of XZF (0, 50, 100, and 200 μg/mL) for 24 h, after which the conditioned medium was collected. This conditioned medium was then placed in the lower compartments of transwell plates to serve as a chemoattractant for prostate cancer cells. To evaluate the migratory capacity of prostate cancer cells, PC-3 (Figure 9A), DU145 (Figure 9C), and LNCaP (Figure 9E) cells were separately seeded into the upper chambers of transwell plates equipped with microporous membrane inserts. Cells were suspended in a serum-free medium to exclude potential confounding effects of serum components, thereby enabling the specific assessment of macrophage-derived factors on cancer cell migration. The results demonstrated a significant, dose-dependent inhibition of migration across all three prostate cancer cell lines when exposed to conditioned medium from XZF-treated THP-1 cells. Notably, conditioned medium from THP-1 cells treated with XZF at concentrations of 50, 100, and 200 μg/mL substantially suppressed the migratory capacity of PC-3 (Figure 9A,B), DU145 (Figure 9C,D), and LNCaP (Figure 9E,F) cells. Quantitative analysis of the migration assay further confirmed that higher XZF concentrations correlated with a more pronounced inhibitory effect on cancer cell migration. This reduction in migratory potential, observed across multiple prostate cancer cell lines, suggests a broad-spectrum anti-migratory effect of XZF.

## 3. Discussion

The XZF formulation, comprising a blend of dried and pulverized fruiting bodies of AC, *Sanghuangporus sanghuang*, *Ganoderma lucidum*, *Ganoderma sinense*, and *Inonotus obliquus*, presents a multifaceted approach in the exploration of anticancer therapeutics. Recent experimental findings have revealed significant differential effects of XZF on prostate cancer cell lines. At concentrations of 100 and 200 μg/mL, XZF showed no triggering effect on PD-L1 expression in either PC-3 or DU145 cell lines. More notably, at 200 μg/mL, XZF demonstrated the ability to counteract IFN-γ-induced PD-L1 expression (10 ng/mL) in both cell lines, suggesting promising therapeutic applications. Cell viability studies using XTT assays revealed distinct patterns of response between the two prostate cancer cell types. PC-3 cells showed significant sensitivity to XZF, with marked inhibitory effects observed across a concentration range of 25–200 μg/mL over a 24–48 h period. In contrast, DU145 cells demonstrated a more limited response, showing only partial inhibition of cell viability at higher XZF concentrations (100–200 μg/mL) and only after 48 h of treatment. AC, endemic to Taiwan, exhibits potential therapeutic activity against a spectrum of cancers, including breast, hepatocellular, and lung cancers. A synergistic apoptotic effect in androgen-refractory PC-3 human prostate cancer cells has been observed when AC is combined with lovastatin, indicating its potential efficacy in prostate cancer management [30]. This synergistic effect aligns with our observed pronounced sensitivity of PC-3 cells to XZF treatment. *Ganoderma lucidum*, another medicinal mushroom in the formulation, is particularly notable for its anti-prostate cancer potential. *Ganoderma lucidum* polysaccharides have been found to impede cell migration in LNCaP prostate cancer cells via the PRMT6 signaling pathway and promote apoptosis in PC-3 prostate cancer cells, suggesting a preventive role in cancer metastasis [31,32]. Furthermore, *Ganoderma lucidum*’s total triterpenoids are reported to arrest the cell cycle and induce apoptosis in prostate cancer cells, augmenting the anti-growth effect of standard prostate cancer treatments such as flutamide and docetaxel [33]. Additional research suggests *Ganoderma lucidum*’s capacity to inhibit prostate cancer cell development and induce apoptosis through the Jak-1/STAT-3 signaling pathway [34]. *Sanghuangporus vaninii*, a traditional medicinal fungus, has demonstrated its anticancer prowess through its polysaccharides, which inhibit cell proliferation, regulate the cell cycle, and induce apoptosis in various cancer cell types, including breast and colorectal cancers [35,36]. *Inonotus obliquus*, commonly known as Chaga mushroom, has shown anticancer properties through autophagy induction in cancer cells, exhibiting activity against melanoma B16-F10 cells and in sarcoma-180 cell-bearing mice [37,38]. The differential response between PC-3 and DU145 cells to XZF treatment suggests that the formulation’s mechanism of action might be dependent on specific cellular characteristics present in PC-3 cells but less pronounced in DU145 cells. While the components of XZF, particularly *AC, Ganoderma lucidum*, and *Sanghuangporus vaninii*, have shown promising results in cancer therapy, specifically for prostate cancer, the effects of *Ganoderma sinense* and *Inonotus obliquus* on prostate cancer require further investigation to fully elucidate their therapeutic potential. Additionally, the observed ability of XZF to modulate PD-L1 expression opens new avenues for investigation into its potential role in cancer immunotherapy.

Numerous medicinal mushrooms within the XZF formulation have demonstrated potential efficacy against prostate cancer, as demonstrated by prior research. *Ganoderma lucidum* has been extensively investigated for its anticancer effects. It has been shown to restrict the proliferation of prostate cancer cells, trigger programmed cell death, and inhibit the formation of new blood vessels [31,39,40]. Extracts of *Ganoderma lucidum* have further been observed to attenuate androgen receptor activity, reduce prostate-specific antigen levels, and influence MAPK and Akt signaling pathways [39,40]. Additionally, the polysaccharides derived from *Ganoderma lucidum* exhibit synergistic interactions with standard therapies for prostate cancer, such as Docetaxel and Flutamide, thereby potentially enhancing their therapeutic outcomes [33]. Another medicinal fungus, AC, has also demonstrated promise in prostate cancer treatment. The compound 4-Acetylantroquinonol B (4AAQB) from AC has been reported to curb prostate cancer advancement by blocking VEGF-driven angiogenesis and metastasis [41]. Furthermore, antrocin, another bioactive constituent of AC, has been found to enhance the sensitivity of prostate cancer cells to radiation therapy by targeting downstream IGF-1R signaling pathways [42]. Although there are no direct reports regarding the effects of *Sanghuangporus sanghuang*, *Ganoderma sinense*, and *Inonotus obliquus* on prostate cancer, studies suggest that Inonotus obliquus may exhibit anticancer potential in other cancer types [43]. Our research has shown that XZF treatment markedly reduced the proportion of PD-1+ CD8+ T-cells, thereby enhancing their cytotoxic activity while leaving PD-1+ CD4+ T-cells largely unaffected. Additionally, XZF was effective in suppressing IFN-γ-induced PD-L1 expression in PC-3 and DU145 prostate cancer cells and inhibited the PD-L1/PD-1 interaction by 93% at a concentration of 200 μg/mL, closely matching the 98% inhibition achieved with atezolizumab. Co-treatment with XZF and atezolizumab completely abolished the PD-L1/PD-1 interaction. Moreover, XZF enhanced the viability of activated CD8+ and CD4+ T-cells in a concentration-dependent manner, likely through its polysaccharide-mediated activation of dendritic cells and promotion of Th1 differentiation. Comprehensive transcriptomic and proteomic investigations are required to elucidate the molecular pathways involved, particularly those associated with PD-1/PD-L1 regulation and adaptive immune responses. Further research is also needed to delineate the individual roles of each component of XZF to clarify their specific mechanisms of action.

Our findings revealed that PC-3 cells exhibited a higher sensitivity to XZF, with pronounced suppression of cell viability observed over a wide concentration range (25–200 μg/mL) within 24–48 h. Conversely, DU145 cells displayed only moderate inhibition, limited to higher concentrations (100–200 μg/mL) after 48 h of exposure. This variation may be attributed to differences in their genetic characteristics and associated signaling mechanisms. PC-3 cells generally exhibit greater sensitivity to treatments compared to DU145 cells due to several factors: PC-3 cells have higher basal and inducible levels of reactive oxygen species, making them more susceptible to oxidative stress-induced cell death [44]. PC-3 cells are AR-negative, while DU145 cells express a mutated AR, which could affect their response to treatments [45,46]. Furthermore, differences in cell cycle regulation, DNA repair mechanisms, and metabolic profiles between the two cell lines could contribute to their varying sensitivities to XZF [47,48]. The greater sensitivity of PC-3 cells might also be related to their higher metastatic potential and more aggressive phenotype, which could make them more vulnerable to certain therapeutic interventions [49]. Further investigation into these molecular differences could provide valuable insights into optimizing XZF-based treatments for different prostate cancer subtypes. For possible molecular mechanisms, PC-3 cells generally exhibit greater sensitivity to treatments compared to DU145 cells due to several factors: PC-3 cells have higher basal and inducible levels of reactive oxygen species, making them more susceptible to oxidative stress-induced cell death [44]. PC-3 cells are AR-negative, while DU145 cells express a mutated AR, which could affect their response to treatments [45,46]. Furthermore, differences in cell cycle regulation, DNA repair mechanisms, and metabolic profiles between the two cell lines could contribute to their varying sensitivities to XZF [47,48]. The greater sensitivity of PC-3 cells might also be related to their higher metastatic potential and more aggressive phenotype, which could make them more vulnerable to certain therapeutic interventions [48]. Further investigation into these molecular differences could provide valuable insights into optimizing XZF-based treatments for different prostate cancer subtypes.

Prostate cancer cells exhibit distinct androgen sensitivity and metabolic profiles, influencing their response to therapeutic agents. PC-3 and DU145 are androgen-insensitive and do not express functional androgen receptors (AR), while LNCaP cells are androgen-sensitive and express functional AR [50]. Previous studies indicate that AR blockade enhances CD8+ T-cell function and improves checkpoint blockade response via increased IFN-γ expression [51]. Metabolically, LNCaP cells display a more oxidative phenotype, whereas PC-3 and DU145 exhibit a glycolytic phenotype [52]. Additionally, PC-3 and DU145 cells express high levels of CD44 (>93%), in contrast to LNCaP cells, which express less than 4% [53]. Growth rates also vary, with PC-3 and DU145 proliferating faster than LNCaP cells [53]. In terms of metastatic potential, PC-3 cells are highly aggressive, DU145 moderately metastatic, and LNCaP lowly metastatic [49]. Our study demonstrated that XZF at 200 μg/mL effectively attenuated IFN-γ-induced PD-L1 expression in PC-3 and DU145 cell lines. XZF exhibited strong inhibition (93%) of the PD-L1/PD-1 interaction, comparable to atezolizumab, with complete blockade achieved when combined. These findings highlight the potential of XZF in immune checkpoint modulation. Further investigations into XZF’s impact on cell viability revealed differential responses among prostate cancer cell lines. XZF significantly suppressed PC-3 cell viability in a dose-dependent manner (25–200 μg/mL) over 24–48 h, suggesting potent antiproliferative activity. DU145 cells exhibited a more restricted response, with inhibition becoming evident only at higher concentrations (100–200 μg/mL) and after prolonged exposure (48 h). Interestingly, LNCaP cells displayed a viability reduction similar to PC-3 cells, suggesting that XZF may exert its effects through AR-independent mechanisms. Beyond direct effects on prostate cancer cells, we explored the role of macrophages in modulating XZF’s activity. Conditioned medium from XZF-treated RAW 264.7 (murine) and THP-1 (human) macrophages differentially influenced prostate cancer cell viability. While a RAW 264.7-derived conditioned medium had limited impact, conditioned medium from THP-1 cells treated with 100–200 μg/mL XZF significantly reduced the viability of PC-3, DU145, and LNCaP cells. These results indicate that XZF may modulate macrophage-secreted factors to exert indirect anticancer effects, with human macrophages showing a greater regulatory impact. Given the crucial role of migration in cancer progression, we also examined XZF’s effect on prostate cancer cell migration in the presence of a macrophage-conditioned medium. Notably, conditioned medium from THP-1 cells treated with 50–200 μg/mL XZF significantly suppressed migration of PC-3, DU145, and LNCaP cells in a dose-dependent manner. This broad-spectrum anti-migratory effect further supports XZF’s potential in limiting prostate cancer progression. Overall, our findings demonstrate that XZF effectively inhibits PD-L1 expression, suppresses cell proliferation, and reduces migration in prostate cancer cells, with both direct and macrophage-mediated effects. Further research is warranted to elucidate XZF’s impact on androgen-sensitive LNCaP cells, particularly in relation to AR signaling and immune interactions. These insights will enhance our understanding of XZF as a potential therapeutic agent for prostate cancer treatment.

Based on a review of the literature, the concentrations of fungal extracts used in cancer cell experiments vary depending on the specific extract and cell line but generally fall within similar ranges to those reported in the attached results. For AC, studies have used concentrations ranging from 50 to 200 μg/mL, with cytotoxic effects observed at higher doses. The 50–200 μg/mL range used in the attached results aligns with these previous findings [54,55,56]. *Sanghuangporus sanghuang* extracts have been tested at 25–350 μg/mL in cancer cells, with growth inhibition seen at higher concentrations [57,58]. The 50–200 μg/mL range in the results is within this established effective range. *Ganoderma lucidum* extracts are commonly used at 100–1000 μg/mL in cancer cell experiments [59,60]. The 50–200 μg/mL concentrations in the results fall on the lower end of this range, likely to assess effects at more moderate doses. For *Ganoderma sinense*, studies have used 50–150 μg/mL concentrations [61,62], similar to the 50–200 μg/mL range in the results. *Inonotus obliquus* extracts are typically tested at 50–300 μg/mL in cancer cells [63,64], which encompasses the 50–200 μg/mL range used in the attached results. Overall, the concentrations used in the experiments align well with ranges established in previous literature and preliminary data for these fungal species, allowing for the assessment of anticancer effects at doses known to be biologically relevant while avoiding excessive toxicity. In addition, XZF demonstrated a favorable safety profile with no significant cytotoxic or immunosuppressive effects in non-tumoral models. Using human embryonic kidney 293T cells as a non-cancerous control model, XZF treatment at concentrations ranging from 50 to 200 μg/mL showed no significant alterations in cellular metabolic activity or viability compared to untreated controls, even during extended 48-h observation periods. Furthermore, rather than exhibiting immunosuppressive properties, XZF actually demonstrated immunostimulatory effects by enhancing the viability of CD8+ and CD4+ T-cells in a dose-dependent manner, supporting T-cell-mediated immune mechanisms. This selective activity profile, where XZF maintains or enhances normal cell viability while specifically targeting cancer cells, suggests it may have potential therapeutic applications with minimal adverse effects on healthy tissues.

The biodistribution and pharmacokinetic profiles of *Antrodia cinnamomea*, *Sanghuangporus sanghuang*, *Ganoderma lucidum*, *Ganoderma sinense*, and *Inonotus obliquus*, key constituents of XZF, have been insufficiently investigated in both animal and human models. For AC, studies in rats have shown that triterpenoids like antcins K and H are the major exposure metabolites, with ergostanes being rapidly absorbed and eliminated while lanostanes persist longer in plasma [65]. A proteoglycan from *Ganoderma lucidum* showed preferential accumulation in the liver and kidneys of mice [66]. *Inonotus obliquus* components, particularly inotodiol, demonstrated good oral bioavailability in rats, with significant distribution to the liver [67]. These findings collectively suggest that the bioactive compounds from XZF generally have good oral bioavailability and tend to accumulate in metabolically active organs, supporting their potential therapeutic effects. However, more comprehensive human studies are needed to fully elucidate their pharmacokinetic profiles and optimize dosing strategies for clinical applications. Based on the available search results, there is limited information specifically addressing the absorption, distribution, metabolism, and excretion (ADME) of AC, *Sanghuangporus sanghuang*, *Ganoderma sinense*, and *Inonotus obliquus*, key constituents of XZF, in animal or human models. However, some relevant information about *Ganoderma lucidum* and its compounds can be summarized. *Ganoderma lucidum*, a medicinal mushroom, produces various pharmacologically active compounds, including triterpenoids and ganoderic acids [68,69]. Ganoderic acids, which are among the main bioactive components of *Ganoderma lucidum*, have been studied for their pharmacokinetics [69]. While specific ADME details are not provided in the search results, it is worth noting that a network pharmacology analysis has been conducted on drug-like compounds from Ganoderma lucidum, suggesting potential applications for chronic inflammatory conditions such as diabetes mellitus [70]. Additionally, some studies have focused on the ADME/T (absorption, distribution, metabolism, excretion, and toxicity) properties of specific compounds from *Ganoderma lucidum*, such as ganoderic acid, a lanosterol triterpene with a wide range of medicinal values [70]. These studies indicate ongoing research interest in understanding the pharmacokinetic properties of XZF and its bioactive compounds, which could potentially inform future investigations into their ADME characteristics in animal or human models.

In this study, we investigated the therapeutic potential and underlying mechanisms of the botanical formulation XZF against prostate cancer, focusing on its direct antiproliferative effects, immunomodulatory properties, and macrophage-mediated anticancer responses. Our findings demonstrated that XZF significantly inhibits the viability and migration of prostate cancer cells, underscoring its direct antitumor efficacy. Furthermore, XZF exhibited notable immunomodulatory activities, characterized by marked downregulation of PD-1 expression in activated CD8^+^ T-cells, suppression of PD-L1 expression in prostate cancer cells, and strong inhibition of the PD-L1/PD-1 interaction, a critical pathway involved in tumor immune evasion. While molecular dynamics (MD) simulations represent a powerful tool for elucidating structure-activity relationships (SAR) and confirming conformational dynamics associated with biological activity, applying such approaches to the current formulation presents significant practical challenges. XZF is a complex botanical mixture comprising multiple extracts, including *Antrodia cinnamomea*, *Sanghuangporus sanghuang*, *Ganoderma lucidum*, *Ganoderma sinense*, and *Inonotus obliquus*. Each of these constituents contains numerous bioactive molecules such as triterpenoids, polysaccharides, and polyphenolic compounds, each potentially contributing to the formulation’s overall pharmacological effects through synergistic interactions. Due to this inherent complexity, accurately selecting a single representative molecular structure suitable for MD simulations becomes problematic, limiting the utility of traditional MD simulation techniques in fully capturing the pharmacological dynamics of the whole formulation. Moreover, bioactivity in botanical formulations like XZF typically emerges from synergistic interactions among multiple components rather than from isolated compounds. For instance, *Antrodia cinnamomea* is known for its immunomodulatory and anticancer triterpenoids and polysaccharides, while *Ganoderma lucidum* possesses similar biological activities driven by both its polysaccharides and triterpenoids. These synergistic and multifactorial interactions further complicate precise computational modeling aimed at defining single-structure interactions. Additionally, observed variations between macrophage responses, notably stronger anticancer effects in human macrophages (THP-1 cells) compared to murine macrophages (RAW 264.7 cells), underscore the complexity inherent in accurately simulating biological responses. Such species-specific differences highlight the challenges in directly translating computational predictions or animal model findings into human therapeutic contexts. Despite these limitations, targeted computational approaches focusing on individual bioactive compounds or specific molecular interactions, such as docking studies or selective MD simulations, could provide valuable mechanistic insights. Future investigations integrating targeted computational methods with experimental validation are warranted and will significantly enhance our understanding of the molecular interactions and mechanisms underpinning XZF’s therapeutic potential in prostate cancer.

Recent research has demonstrated that the medicinal mushrooms of XZF may promote cytotoxic T-cell infiltration into the tumor microenvironment, thereby enhancing antitumor immune responses. *Ganoderma lucidum*’s polysaccharide (GLP) has been shown to enhance antitumor immunity and boost the effectiveness of anti-PD-1 immunotherapy in colorectal cancer by increasing beneficial T-cells while decreasing immunosuppressive cells, along with improving gut microbiome health and metabolic factors. The research highlights GLP’s ability to increase the proportion of cytotoxic CD8+ T-cells and Th1 helper cells in both spleen and tumor tissues while simultaneously alleviating microbiota dysbiosis and improving metabolic markers like short-chain fatty acid production [71]. Similarly, AC has exhibited remarkable properties in regulating immune response during radiation therapy without providing any protective effects to tumor cells. Instead, it enhances radiation-induced inflammation and cytotoxicity in cancer cells, suggesting its potential as a complementary cancer treatment [72]. What makes these findings particularly noteworthy is that both mushrooms work by modulating the immune system’s response to cancer, albeit through different mechanisms. This dual-action approach, combining traditional medicinal mushrooms with modern cancer treatments, indicates they could be valuable additions to existing cancer therapies, particularly when used in conjunction with immunotherapy or radiation therapy.

AC, primarily studied for its effects on immune cells, has shown the capability to inhibit crucial immunoregulatory signaling pathways. This suggests a role in attenuating over-activated cytotoxic and inflammatory responses, particularly valuable in radiation-induced tissue damage scenarios. Notably, AC enhances radiation-induced inflammation and cytotoxicity in cancer cells rather than providing radioprotective effects [72]. Its water extract, CCM111, demonstrates significant inhibition of the NF-κB and STAT3 signaling pathways in HEK293 cells, suggesting a suppressive influence on these pathways integral to inflammation and immune responses [73]. *Ganoderma sinense*, closely related to *Ganoderma lucidum*, exhibits notable immunomodulatory activities. Its polysaccharide-enriched fraction, especially from the stipe part, markedly stimulates human peripheral blood mononuclear cell proliferation and enhances cytokine production, suggesting benefits for immunosuppressed patients [74]. *Ganoderma lucidum* extract is another focus of study, showing a significant boost to immune system function and antitumor immunostimulatory activity. This extract appears to have minimal toxic effects on liver and kidney functions in tumor-bearing mice and enhances the immune response by increasing white blood cell and lymphocyte counts, including natural killer cells and T-cells (CD4+ and CD8+ cells) [75]. *Inonotus obliquus* and *Sanghuangporus sanghuang*, however, have limited specific research on their direct impact on immune cells in the context of prostate cancer. Studies indicate anti-inflammatory and anticancer properties, but more research is needed for a comprehensive understanding of its role in modulating immune responses in prostate cancer [76]. In XZF, AC may play a pivotal role, notably in modulating T-cell activity. High-molecular-weight polysaccharides in AC are crucial in enhancing dendritic cell functionality and activating antigen-specific T-cells, promoting Th1 differentiation [21]. XZF, at specific concentrations, enhances the viability of activated CD8+ and CD4+ T-cells, suggesting its utility in T-cell-mediated immune responses. The study also extends to the viability of macrophages and prostate cancer cells, demonstrating the partial inhibitory effect of the medium conditioned by RAW 264.7 cells treated with XZF on the viability of PC-3 cells, indicating a potential role in immunotherapy for prostate cancer.

In the evolving field of cancer immunotherapy, particularly regarding prostate cancer, the exploration of medicinal mushrooms like AC, *Sanghuangporus sanghuang, Ganoderma lucidum, Ganoderma sinense*, and *Inonotus obliquus* offers a promising yet under-researched avenue, specifically in their interactions with the PD-L1 and PD-1 pathways. These fungi are rich in bioactive compounds, including polysaccharides, triterpenoids, and phenolic compounds, known for immune modulation, antioxidant activities, and direct antitumor effects. However, their roles in affecting the PD-1/PD-L1 pathways, which are crucial in cancer immune evasion and immunotherapy, are not fully understood. In particular, the lack of specific research on AC’s effects on the PD-L1 and PD-1 pathways highlights a gap in understanding the immunotherapeutic potential of this mushroom. Known for its bioactive compounds that offer immune modulation and anticancer activities, further investigation into how it influences these crucial cancer checkpoints could be insightful. Similarly, *Sanghuangporus sanghuang*, with its rich composition of bioactive compounds, has shown potential in tumor growth regulation and immune response. However, its interaction with PD-L1 and PD-1 pathways needs more exploration, given their critical role in the tumor microenvironment. *Ganoderma lucidum*, with its history in traditional medicine and array of bioactive molecules, also presents an unexplored area in its impact on the PD-1/PD-L1 immune checkpoint pathway. This research could be pivotal in assessing its adjunct potential in cancer treatment, particularly in enhancing the efficacy of PD-1/PD-L1 inhibitors. *Ganoderma lucidum*, with its traditional medicine background, warrants investigation for potential enhancement of PD-1/PD-L1 inhibitor efficacy [77]. For *Inonotus obliquus*, known for its bioactive compounds that modulate immune responses and impact tumor growth, the direct interactions with PD-L1 and PD-1 checkpoints are yet to be elucidated. Prostate cancer is characterized by low T-cell infiltration, rendering it less responsive to conventional immunotherapies. Our study investigated whether XZF could enhance T-cell viability and modulate immune responses, particularly in CD8+ and CD4+ T-cells. Research demonstrates that XZF significantly reduces PD-1+ CD8+ T-cell populations and disrupts PD-L1/PD-1 expression and interaction. Furthermore, XZF inhibits IFN-γ-induced PD-L1 expression in PC-3 cells, effectively preventing the PD-L1/PD-1 interaction. These effects suggest XZF’s potential role in modulating key immunosuppressive pathways in prostate cancer. Notably, XZF treatment significantly decreases PD-1 expression in CD8^+^ T-cells, which may alleviate T-cell exhaustion and enhance antitumor activity, thereby improving the immune microenvironment within prostate cancer tissue. Enhanced proliferation and viability of CD8+ and CD4+ T-cell subsets further support XZF’s immunomodulatory potential. These findings highlight the need for more focused research on XZF’s influence on PD-1/PD-L1 pathways, offering valuable insights for developing novel immunotherapeutic strategies for prostate cancer.

Our findings demonstrated that XZF significantly impacts the PD-1/PD-L1 immune checkpoint pathway, with 93% inhibition of PD-1/PD-L1 interaction at 200 μg/mL and complete blockade when combined with atezolizumab—particularly relevant since acquired resistance often involves immune checkpoint upregulation. XZF enhanced both CD8+ and CD4+ T-cell viability while reducing PD-1+ CD8+ T-cell populations, suggesting it could help maintain T-cell functionality despite chronic antigen exposure that typically causes exhaustion and therapy resistance. Additionally, XZF modulated macrophage-mediated responses through human THP-1 cells, potentially reprogramming the tumor microenvironment toward a more immunogenic state. However, important limitations include the lack of direct resistance model testing, differential responses between PC-3 and DU145 cells, and the absence of in vivo validation. Future studies should examine XZF in acquired resistance models to PD-1/PD-L1 inhibitors, investigate its effects on other immune checkpoints, assess long-term T-cell function impacts, and evaluate its influence on resistant tumor microenvironments. Additional focused research is needed to definitively establish its role in overcoming acquired immunotherapy resistance in prostate cancer.

Macrophages play a pivotal role in cancer progression, modulating therapy resistance, immune checkpoint blockade therapies, and the tumor microenvironment. The interaction between macrophages and bioactive compounds derived from medicinal mushrooms has been increasingly investigated due to their immunomodulatory and anticancer properties [78,79,80,81]. XZF, derived from medicinal mushrooms, has shown significant effects on macrophage-mediated cancer cell responses. In prostate cancer models, conditioned media from RAW 264.7 and THP-1 macrophages treated with XZF at concentrations of 100 and 200 μg/mL partially suppressed the viability of PC-3 and DU145 prostate cancer cells. Interestingly, the inhibitory effects were more pronounced in THP-1-derived macrophages, suggesting a species-specific difference in macrophage response to XZF treatment. Furthermore, conditioned media from XZF-treated THP-1 cells significantly inhibited prostate cancer cell migration in a dose-dependent manner, as demonstrated in transwell assays. This suppression of both cell viability and migration underscores the modulatory capacity of XZF on macrophage-secreted factors, which subsequently influence prostate cancer progression. Parallel findings from studies on medicinal mushrooms, including AC, *Sanghuangporus sanghuang*, *Ganoderma lucidum*, *Ganoderma sinense*, and *Inonotus obliquus*, emphasize their immunomodulatory effects via macrophage activation. ZnF3 from AC induces apoptosis in lung cancer cells and activates macrophages via the AKT/mTOR pathway [82]. Similarly, *Sanghuangporus sanghuang* extract can induce the production of pro-inflammatory cytokines in macrophages [83]. *Ganoderma lucidum* polysaccharides (GLPS) regulate macrophage polarization, promoting an M1 phenotype through MAPK/NF-κB signaling [84]. This macrophage polarization enhances antitumor activity, demonstrating the therapeutic potential of fungal-derived compounds. Moreover, *Ganoderma sinense*’s polysaccharide GSP-2 specifically activates TLR4 on macrophages, inducing cytokine secretion and immune modulation [85]. Additionally, *Inonotus obliquus* polysaccharides, such as AcF1 and AcF3, act as strong agonists for TLR2 and TLR4, enhancing nitric oxide production and cytokine secretion, which contribute to tumoricidal macrophage activity [86]. These interactions collectively contribute to an antitumor microenvironment, underscoring the potential of medicinal mushrooms in cancer immunotherapy [78,79,80,81]. The convergence of findings from both studies suggests a shared mechanistic pathway where fungal bioactive compounds modulate macrophage phenotypes, cytokine secretion, and downstream effects on cancer cells. Specifically, XZF treatment and polysaccharides from medicinal mushrooms share similar outcomes, including macrophage activation and suppression of cancer cell viability and migration. The observed species-specific effects in RAW 264.7 and THP-1 macrophages suggest that human macrophages might respond more robustly to fungal-derived compounds, emphasizing the translational potential of these findings. In conclusion, the immunomodulatory effects of XZF on macrophage-mediated prostate cancer suppression align with findings from studies on medicinal mushroom polysaccharides. These results emphasize the translational potential of medicinal mushroom-derived compounds as adjunct therapies in cancer treatment, particularly in targeting macrophage-mediated pathways. Future studies should focus on elucidating the molecular underpinnings of macrophage-tumor cell interactions in response to fungal bioactives and optimizing therapeutic formulations for clinical applications.

Emerging evidence indicates that specific medicinal mushroom components of XZF may play a role in overcoming cancer resistance to immunotherapeutic interventions. AC has shown promise in enhancing the sensitivity of cancer cells to chemotherapy and radiotherapy [42,87]. It has been found to inhibit cancer stem cells, which are often associated with drug resistance and tumor recurrence [87,88]. *Ganoderma lucidum* extract has demonstrated the ability to reverse multidrug resistance in breast cancer cells by inhibiting the ATPase activity of P-glycoprotein [89]. *Ganoderma lucidum* extract has also been shown to promote tumor cell pyroptosis and enhance antitumor immune responses [90]. *Inonotus obliquus*, commonly known as Chaga, has exhibited anticancer properties against various types of cancer, including bladder cancer [91]. It has been found to inhibit cancer stem cell markers and enhance the effects of chemotherapy [91]. While specific studies on *Sanghuangporus sanghuang* and *Ganoderma sinense* in the context of immunotherapy resistance were not prominent in the search results, the overall body of research suggests that these medicinal mushrooms and their bioactive compounds have the potential to overcome drug resistance and enhance the efficacy of conventional cancer treatments [92,93,94]. These findings indicate that further research into the use of XZF as adjuvants to immunotherapy could yield promising results in addressing cancer resistance.

## 4. Materials and Methods

### 4.1. Cell Culture and Treatment

The following cell lines were obtained from the Bioresource Collection and Research Center (Taiwan): 293T cell line (human embryonic kidney), PC-3 cell line, DU-145 cell line, and LNCaP cell line (human prostate cancer), RAW 264.7 cell line (mouse macrophage), and THP-1 cell line (human macrophage). PC-3 cell line, DU-145 cell line, and RAW 264.7 cell line were maintained in Dulbecco’s Modified Eagle’s medium (DMEM; Invitrogen Corp., Carlsbad, CA, USA) supplemented with 10% fetal bovine serum (FBS). LNCaP cells and THP-1 cells were cultured in the RPMI 1640 medium (Invitrogen Corp., Carlsbad, CA, USA) with 10% FBS. All cultures were maintained at 37 °C in a humidified atmosphere containing 5% CO_2_. *Antrodia cinnamomea* formula (from XIANZHIFANG-HERBAL FIVE ELEMENTS CAPSULES^®^, XZF) was procured from Cheng-Feng Biotechnology Co., Ltd. (Taichung, Taiwan). Antrodia cinnamomea formula (XZF) is a composite mixture, derived from equimolar proportions (1:1:1:1:1 by weight) of dried and pulverized fruiting bodies of *Antrodia cinnamomea*, *Sanghuangporus sanghuang*, *Ganoderma lucidum*, *Ganoderma sinense*, and *Inonotus obliquus*. For the extraction process, 250 g of the powdered fruiting bodies were immersed in 500 milliliters of ethanol at ambient temperature for three days. Subsequent to the initial extraction, the solution was filtered using filter paper, and the remaining solid material was subjected to two additional extraction cycles under identical conditions. The filtrates from these three extraction phases were amalgamated and then subjected to evaporation under reduced pressure to yield a dry mass of 20 g. The resulting extract was redissolved in ethanol and preserved at a temperature of −20 °C for future use. In the experimental assays, varying concentrations of the XZF were prepared by diluting the ethanol-based stock solution. Recombinant IFN-γ (Cell Guidance Systems, Cambridge, UK; accession number: P01579) was employed for treatments. Cells were grown to 60–70% confluence before exposure to drug solutions prepared in water at specified concentrations. Control groups received vehicle (water) only.

### 4.2. Quality Control of XZF

Quality control of XZF was conducted by HPLC according to a previous study performed by the OHC lab in Chia Nan University of Pharmacy & Science Department of Food Science & Technology [95]. In brief, their triterpenoid patterns of XZF were analyzed by reverse phase HPLC using a gradient elution of acetonitrile/2% acetic acid (1/4 and 1/2). The reference standard is the purified extract of *Antrodia cinnamomea* fruiting bodies, with triterpenoids used as standard compounds (the retention time of components in the HPLC chromatogram is 50–100 min, including Antcin A, B, C, K, and Zhankuic acid B, C, Dehydroeburicoic acid and 4,7-dimethoxy-5-methyl-1,3-benzodioxole).

### 4.3. Cell Viability Assessment

Cell viability was determined using the XTT (2,3-Bis-(2-Methoxy-4-Nitro-5-Sulfophenyl)-2H-Tetrazolium-5-Carboxanilide) assay as previously described [96]. Cells were seeded in 96-well plates (1 × 10^3^ cells/well) in a medium containing 10% FBS. After attachment, the medium was replaced, and treatments were administered at specified concentrations and durations. Cell viability was measured using an XTT assay kit (Biological Industries, Israel; catalog number: 20-300-1000) by quantifying formazan complex formation at 492 nm with an ELISA reader (Bio-Rad Laboratories, Inc., Hercules, CA, USA).

### 4.4. Isolation of Splenic CD4^+^ and CD8^+^ T-Cells and T-Cell Activation Assays

The experimental protocols were approved by the Animal Care and Use Committee of Chang Gung Memorial Hospital (Approval No. 201922403). All procedures complied with national guidelines set forth by the National Laboratory Animal Center in Taiwan. Ten-week-old BALB/c mice (18–20 g) were obtained from BioLASCO Taiwan Co., Ltd. (Taipei, Taiwan). Following euthanasia via carbon dioxide inhalation, spleens were aseptically collected. Spleens were dissociated into single-cell suspensions using a cell strainer. Phosphate-buffered saline (PBS) was used to rinse the culture dishes, and cells were harvested by centrifugation, discarding the supernatant. Red blood cells were lysed by resuspending the pellet in 5 mL of lysis buffer for 2 min at room temperature, followed by dilution with 45 mL of Gibco Roswell Park Memorial Institute (RPMI) medium or PBS. The cell pellet was resuspended in a buffer composed of PBS (pH 7.2), 2.5% bovine serum albumin (BSA), and 10 mM ethylenediaminetetraacetic acid (EDTA). Biotin-antibody cocktails from the MojoSort™ Mouse CD4+ T-Cell Isolation Kit (BioLegend, San Diego, CA, USA, Catalog No. 480006) or CD8+ T-Cell Isolation Kit (BioLegend, San Diego, CA, USA, Catalog No. 480008) were added to the buffer. Streptavidin-coated nanobeads were incorporated into the mixture, which was incubated on ice for 15 min. Magnetic separation was performed by placing the tube on a magnet for 5 min to isolate the targeted cells. The isolated T-cells were subsequently washed, centrifuged, and collected for downstream assays.

### 4.5. Flow Cytometry Analysis of T-Cell Subpopulations

A total of 1 × 10^6^ cells were seeded into 100-mm plates and incubated overnight. After the designated treatments were applied for specified durations, the culture medium was removed, and the treated cells were collected. The supernatant was discarded following centrifugation, and the cell pellet was resuspended in preparation for staining. Specific antibodies were used to target surface markers, including FITC-conjugated anti-mouse CD279 (PD-1) antibody (clone 29F.1A12, BioLegend, San Diego, CA, USA, catalog no. 135214, dilution 1:100), APC-conjugated anti-mouse CD8a antibody (clone 53-6.7, Becton, Dickinson and Company, catalog no. 561093, dilution 1:100), and APC-conjugated anti-mouse CD4 antibody (clone RM4-5, Becton, Dickinson and Company, catalog no. 561091, dilution 1:100). Flow cytometry was performed using a BD FACSCanto cytometer (Becton, Dickinson and Company, Franklin Lakes, NJ, USA) to analyze the expression of surface markers on T-cell subpopulations.

### 4.6. PD-1/PD-L1 Homogeneous Analysis

The PD-1/PD-L1 interaction was assessed using a homogeneous binding assay kit (BPS Bioscience Inc., San Diego, CA, USA; catalog number: 72014) as previously described [97]. Test compounds were incubated with PD-1 and biotinylated PD-L1 for 60 min, followed by the addition of acceptor and donor beads for Alpha-count measurements.

### 4.7. Western Blot Analysis

Protein expression was analyzed by Western blot as previously described [98]. Cell lysates were prepared from treated (24 h) and control samples according to the manufacturer’s protocol. Proteins were resolved on 6% or 12% SDS-PAGE gels and transferred to PVDF membranes. Membranes were blocked with 5% nonfat dried milk (30 min) and incubated with primary antibodies (3 h, room temperature) in 1% nonfat milk/TBST solution. Primary antibodies included anti-PD-L1 (clone: D4H1Z, Cell Signaling Technology, Inc., Essex County, MA, USA; 1:1000) and anti-Vinculin (Sigma, USA; catalog number: V9131; 1:30,000). After TBST washing, membranes were incubated with HRP-conjugated secondary antibodies (1 h, room temperature): anti-mouse (Cell Signaling Technology, Inc.; catalog number: 7076; 1:10,000) or anti-rabbit (Cell Signaling Technology, Inc., MA, USA; catalog number: 7074; 1:10,000). Protein signals were visualized using Super Signal chemiluminescent substrate (Pierce Biotechnology Inc., Rockfort, IL, USA; catalog number: 34087).

### 4.8. Protein Quantification Analysis

Western blot band intensities were quantified using AlphaEase^®^FC software, version 4.0.0, following the manufacturer’s guidelines. Background signal was systematically removed to ensure accurate measurement of protein expression levels. The untreated control group served as the reference standard, and relative protein expression levels in treated groups were determined by calculating the ratio of their band densities to that of the control.

### 4.9. Migration Analysis

Cell migration was evaluated using a transwell assay as previously described [97]. RAW 264.7 or THP-1 cells (1 × 10^5^ cells/well) were cultured with or without test compounds for 24 h, and the resulting conditioned media were collected in lower chambers. PC-3 or DU145 cells (1 × 10^5^ cells/well) were seeded in serum-free medium in upper chambers and allowed to migrate for 24 h. Migrated cells were fixed, stained with 1% toluidine blue, and quantified by counting six random fields per well. All conditions were tested in triplicate with a minimum of two independent experiments.

### 4.10. Statistical Analysis

Experimental data are presented as mean ± standard error of the mean (*n* = 3–6 replicates) from a minimum of three independent experiments. Statistical comparisons between the two groups were performed using an unpaired two-tailed Student’s *t*-test. Multiple group comparisons utilized one-way ANOVA followed by Tukey’s post-hoc test. Statistical significance was set at *p* < 0.01. Analyses were conducted using SPSS software (version 13.0; SPSS Inc., Chicago, IL, USA).

## 5. Conclusions

In conclusion, our findings reinforce the remarkable selectivity of XZF, demonstrating minimal cytotoxicity toward non-cancerous 293T cells while significantly enhancing the viability of both CD4+ and CD8+ T lymphocytes in a dose-dependent manner. XZF exhibited potent immunomodulatory properties by selectively reducing PD-1 expression in CD8+ T-cells and achieving 93% inhibition of PD-L1/PD-1 interaction at 200 μg/mL, with complete blockade when combined with atezolizumab. Additionally, XZF displayed direct anticancer effects across multiple prostate cancer cell lines, showing a pronounced antiproliferative response in PC-3 and LNCaP cells over a broad concentration range (50–200 μg/mL), whereas DU145 cells exhibited partial sensitivity only at higher concentrations (100–200 μg/mL). Notably, our investigation further elucidated a novel macrophage-mediated mechanism by which conditioned medium from XZF-treated human THP-1 macrophages significantly suppressed both the viability and migratory capacity of prostate cancer cells. This effect was particularly pronounced compared to murine macrophage models, underscoring the potential clinical relevance of XZF’s immunomodulatory effects. Collectively, these results suggest that XZF not only exerts direct anticancer activity but also modulates macrophage-derived factors to indirectly influence prostate cancer cell survival and migration. Given its selective immune checkpoint inhibition, direct cytotoxicity, and macrophage-mediated antitumor effects, XZF emerges as a promising therapeutic candidate for prostate cancer treatment, particularly in combination with existing immunotherapies.

## Figures and Tables

**Figure 1 ijms-26-02684-f001:**
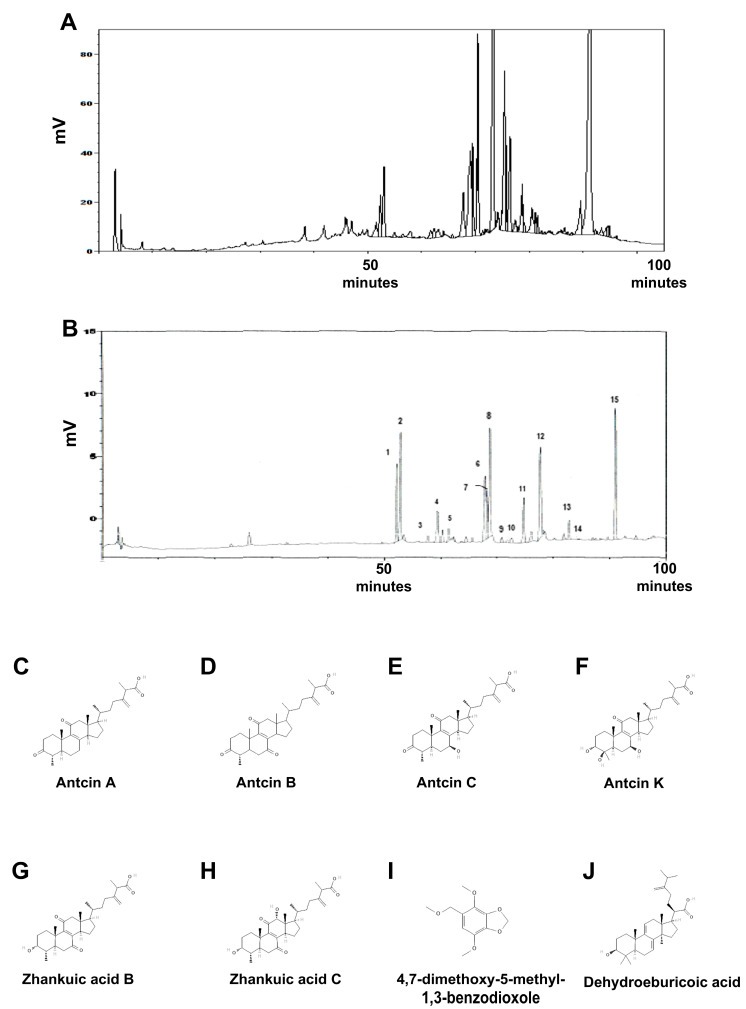
The fingerprint chromatography of XZF by HPLC analysis. (**A**) The HPLC fingerprint chromatography of XZF. (**B**) The chromatogram of the HPLC profile for the extract of a 3-year-old *Antrodia cinnamomea* fruiting body (used as a reference standard). It indicates the identification of various peaks, including: 1: Unknown; 2: Antcin K; 3: Unknown; 4: Antcin H; 5: 4,7-dimethoxy-5-methyl-1,3-benzodioxole; 6: Antcin C; 7: Unknown; 8: Zhankuic acid; 9: Antcin B; 10: Antcin A; 11: Dehydrasulphurenic acid; 12: Zhankuic acid B; 13: Zhankuic acid A; 14: Unknown; 15: Dehydroeburicoic acid. The structure of Antcin A (**C**), B (**D**), C (**E**), K (**F**), Zhankuic acid B (**G**), C (**H**), 4,7-dimethoxy-5-methyl-1,3-benzodioxole (**I**), and Dehydrasulphurenic acid (**J**).

**Figure 2 ijms-26-02684-f002:**
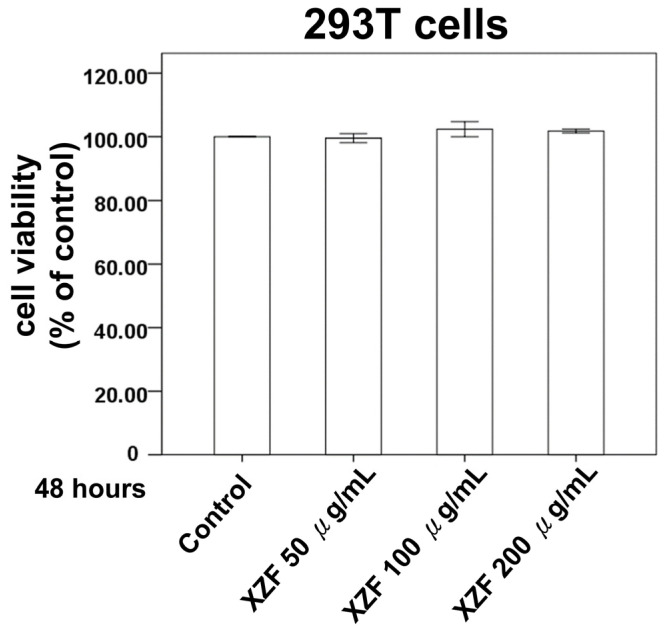
The impact of specified treatments on the cell viability of 293T cells. 293T cells underwent treatment with either ethanol alone or 50, 100, or 200 μg/mL XZF for a duration of 24 h. The viability of cells was assessed using the XTT assay. The presented results are representative of a minimum of three independent experiments. (Error bars = mean ± S.E.M.).

**Figure 3 ijms-26-02684-f003:**
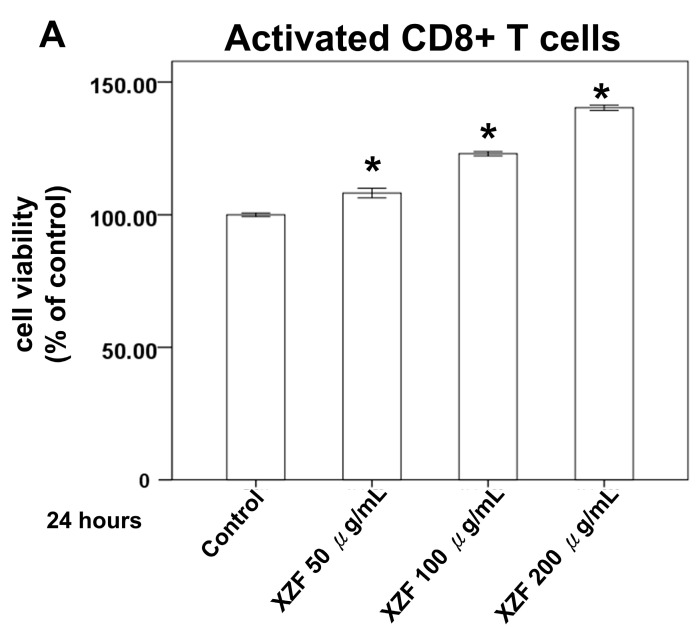
Impact of XZF treatment on T-cell viability. (**A**) Activated CD8+ T-cells were treated with ethanol (control) or XZF (50, 100, 200 μg/mL) for 24 h, followed by XTT assay assessment of cell viability; (**B**) Activated CD4+ T-cells were treated with ethanol (control) or XZF (50, 100, 200 μg/mL) for 24 h, followed by XTT viability assessment. Data represent mean ± S.E.M. from three independent experiments. * *p* < 0.01 vs. control.

**Figure 4 ijms-26-02684-f004:**
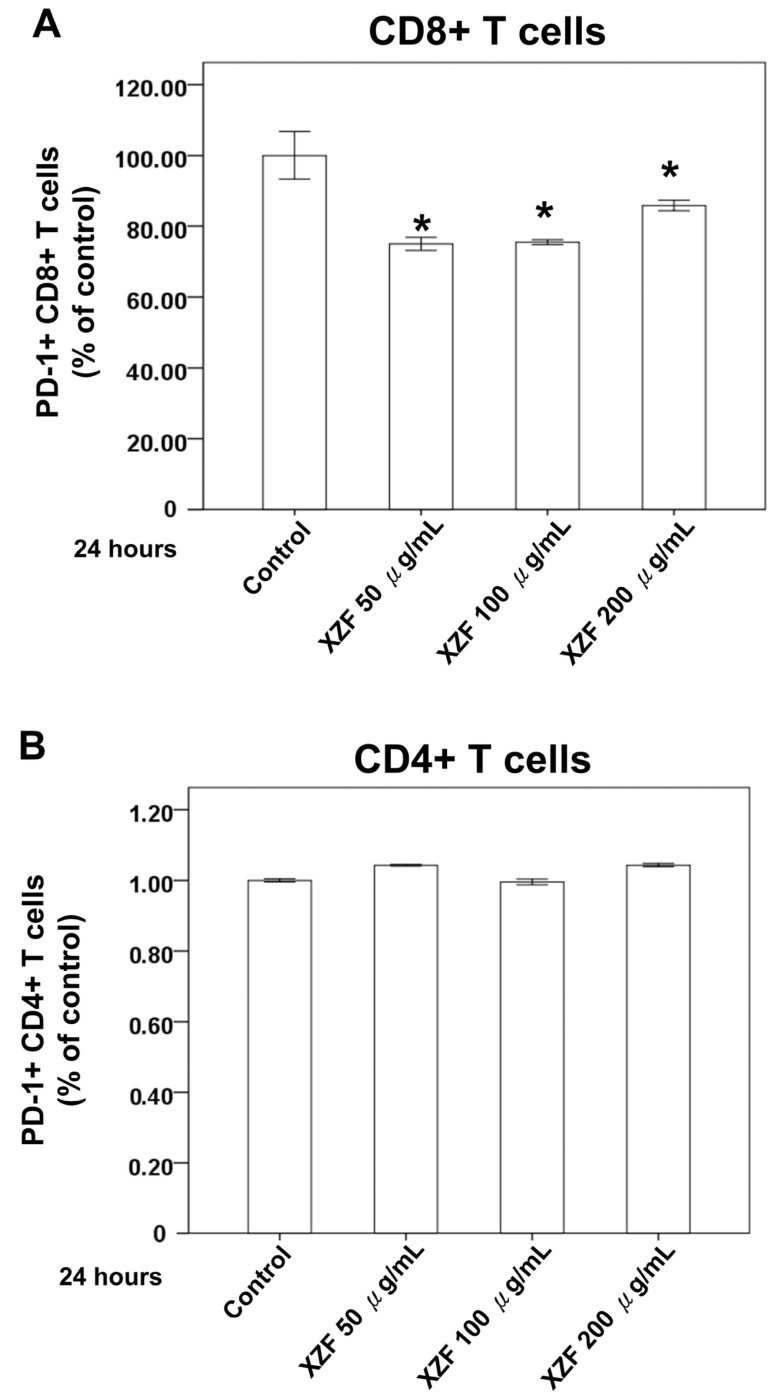
Effect of XZF on T-cell PD-1 expression. (**A**) Inactive CD8+ T-cells were treated with ethanol (control) or XZF (50, 100, 200 μg/mL) for 24 h, followed by flow cytometric analysis of PD-1 expression. Bar graphs show quantification of PD-1+ CD8+ T-cell populations; (**B**) Inactive CD4+ T-cells underwent identical treatment and analysis for PD-1 expression, with bar graphs showing PD-1+ CD4+ T-cell quantification. Data represent mean ± S.E.M. from three independent experiments. * *p* < 0.01 vs. control.

**Figure 5 ijms-26-02684-f005:**
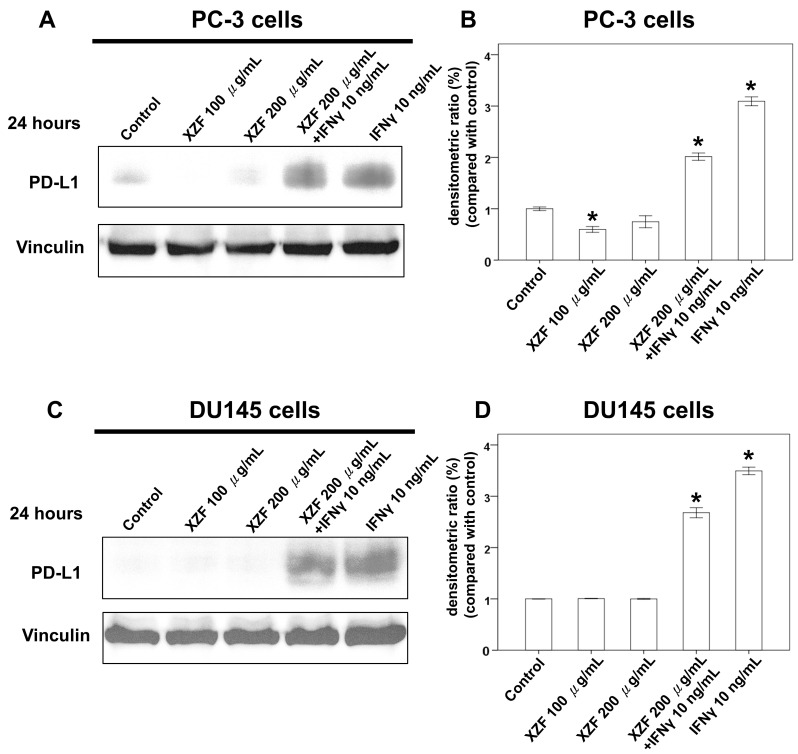
Effects of XZF on PD-L1 expression and PD-L1/PD-1 interaction in prostate cancer cells. (**A**) PC-3 cells were treated with ethanol (control), XZF (100 or 200 μg/mL), IFN-γ (10 ng/mL), or IFN-γ combined with XZF (200 μg/mL) for 24 h, followed by total protein extraction. (**B**) Densitometric analysis of PD-L1 normalized to Vinculin in PC-3 cells, expressed as fold change relative to the control. (**C**) DU145 cells were subjected to the same treatment conditions, followed by protein extraction. (**D**) Densitometric analysis of PD-L1 normalized to Vinculin in DU145 cells, expressed as fold change relative to the control. (**E**) PD-L1/PD-1 interaction was evaluated using a homogeneous assay following treatment with ethanol (control), XZF (200 μg/mL), atezolizumab (1 μg/mL), or a combination of atezolizumab and XZF. Data are presented as mean ± S.E.M. from three independent experiments. * *p* < 0.01 vs. control.

**Figure 6 ijms-26-02684-f006:**
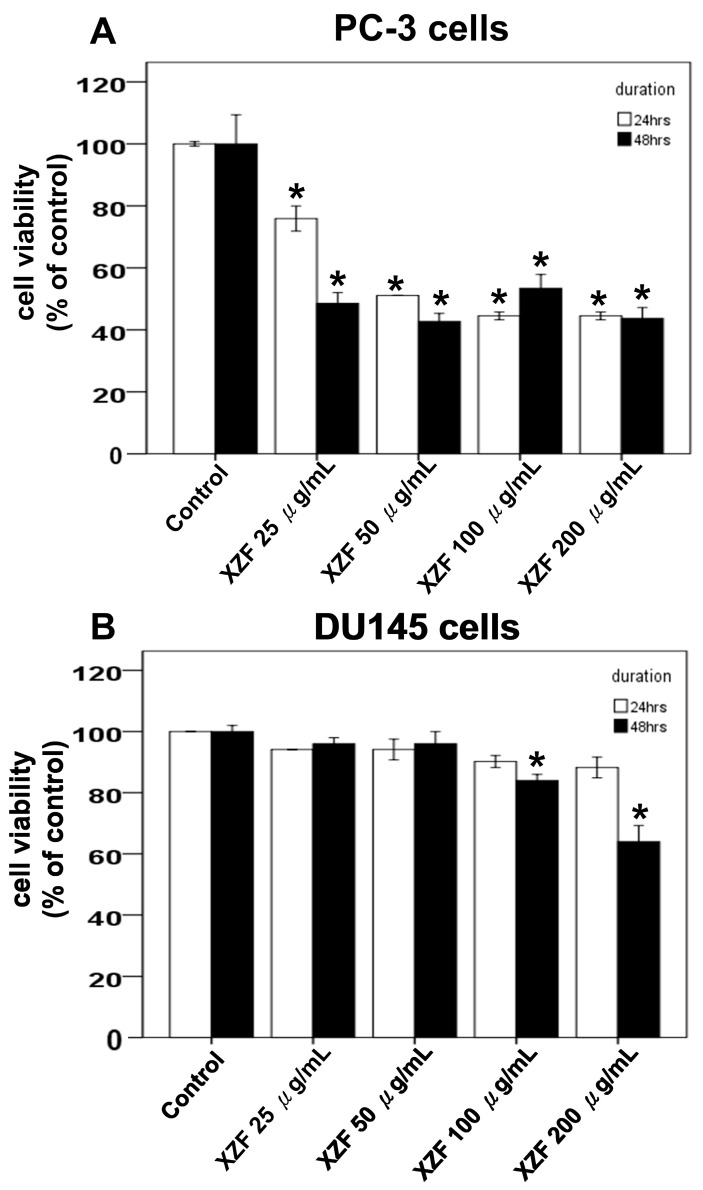
Effects of XZF on Prostate Cancer Cell Viability. (**A**) PC-3 cells were exposed to ethanol (control) or XZF at concentrations of 50, 100, and 200 μg/mL for 24 and 48 h, followed by assessment of cell viability using the XTT assay. (**B**) DU145 cells were subjected to the same treatment and viability evaluation. (**C**) LNCaP cells were similarly treated and analyzed for viability. Data are presented as mean ± S.E.M. from three independent experiments. * *p* < 0.01 compared to the control group.

**Figure 7 ijms-26-02684-f007:**
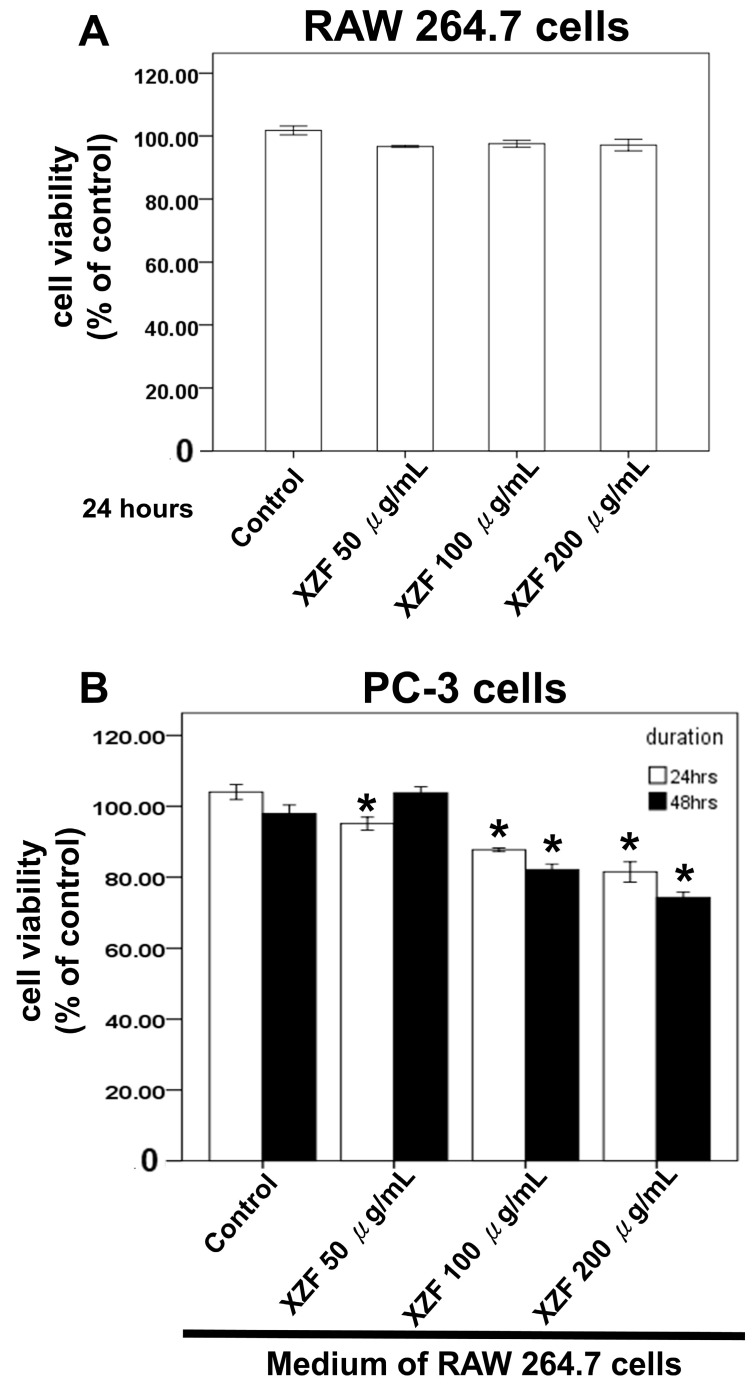
XZF effects on RAW 264.7 and PC-3 cells. (**A**) RAW 264.7 cell viability was assessed by XTT assay after 24-h treatment with ethanol (control) or XZF (100, 200, 300 μg/mL); (**B**) PC-3 cells were cultured in conditioned medium from treated RAW 264.7 cells for 24–48 h, followed by XTT viability assessment. Data represent mean ± S.E.M. from three independent experiments. * *p* < 0.01 vs. control.

**Figure 8 ijms-26-02684-f008:**
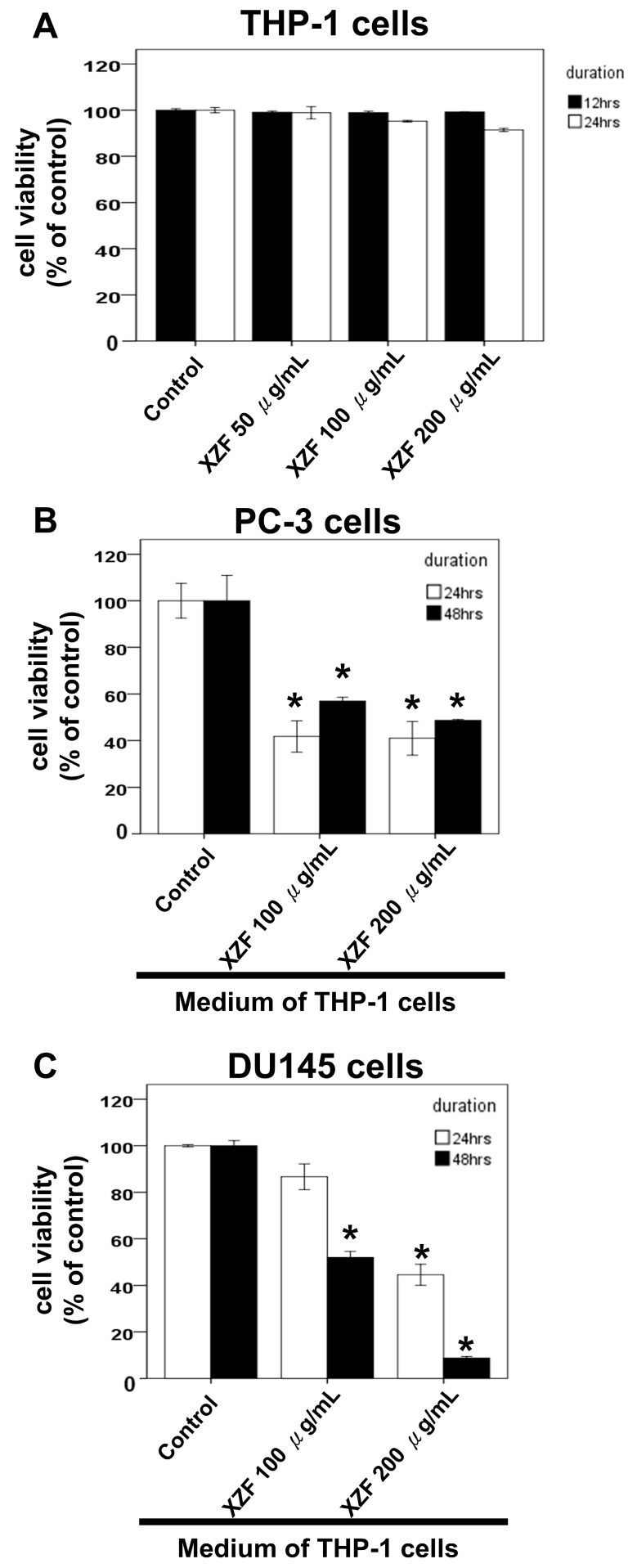
Effects of XZF on THP-1 and Prostate Cancer Cell Viability. (**A**) Viability of THP-1 cells was assessed following 24- and 48-h treatment with ethanol (control) or XZF at concentrations of 100, 200, and 300 μg/mL. (**B**) PC-3 cells and (**C**) DU-145 cells were cultured in a conditioned medium obtained from XZF-treated THP-1 cells for 24 or 48 h, after which cell viability was evaluated. (**D**) LNCaP cells were cultured in a conditioned medium from XZF-treated THP-1 cells for 24 h, followed by viability assessment. Data are expressed as mean ± S.E.M. from three independent experiments. * *p* < 0.01 vs. control.

**Figure 9 ijms-26-02684-f009:**
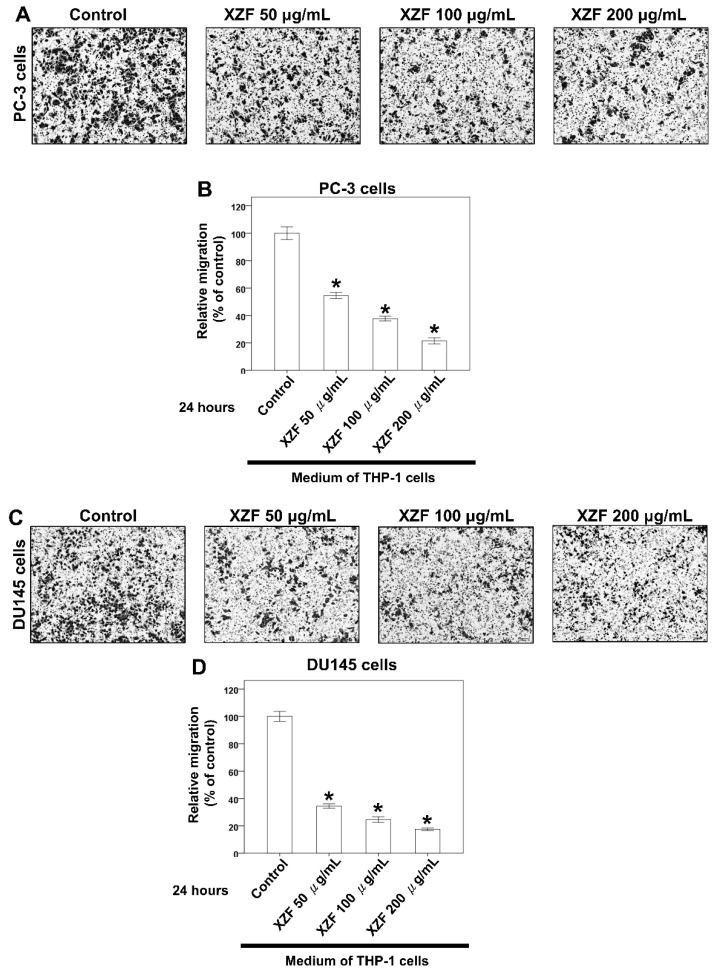
Effects of XZF on Prostate Cancer Cell Migration. (**A**) PC-3 cell migration was evaluated using a transwell assay, with conditioned medium from XZF-treated THP-1 cells placed in the lower chamber. (**B**) Quantification of PC-3 cell migration. (**C**) DU145 cell migration assessed under the same experimental conditions. (**D**) Quantification of DU145 cell migration. (**E**) LNCaP cell migration evaluated under identical conditions. (**F**) Quantification of LNCaP cell migration. Cells were allowed to migrate for 24 h before imaging at 100× magnification and subsequent quantification. Data are presented as mean ± S.E.M. from three independent experiments, normalized to the control group. * *p* < 0.01 vs. control.

## Data Availability

All data generated or analyzed during this study are indicated in this article (with no patient data). The datasets generated during and/or analyzed during the current study are available from the corresponding author upon reasonable request.

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
