# Peer review of "Antrodia cinnamomea Formula Suppresses Prostate Cancer Progression via Immune Modulation and PD-1/PD-L1 Pathway Inhibition"

_ijms, 2025, doi:10.3390/ijms26062684_

Round 1
Reviewer 1 Report (Previous Reviewer 2)
Comments and Suggestions for Authors
I thank the authors for addressing all the concerns raised. However, I suggest including an androgen-sensitive cell line, such as LNCaP, in the study. Additionally, I recommend citing the following paper in the discussion: https://doi.org/10.1038/s41586-022-04522-6.
Author Response
Response to Reviewer 1:
We sincerely thank Reviewer 1 for their constructive feedback, which has helped us to improve the quality of our manuscript. Below, we address each comment point-by-point:
1. I thank the authors for addressing all the concerns raised. However, I suggest including an androgen-sensitive cell line, such as LNCaP, in the study. Additionally, I recommend citing the following paper in the discussion: https://doi.org/10.1038/s41586-022-04522-6.
Response: We appreciate the reviewer’s insightful suggestion regarding the use of androgen-sensitive cell lines, such as LNCaP cells, for more comprehensive results. However, due to the limited 7-day revision period, conducting experiments using LNCaP cells is not feasible at this time. To address this, we plan to incorporate androgen-sensitive cell lines in future research to further substantiate our findings. Additionally, we have revised the discussion section in the manuscript to reflect this feedback:
“PC3 and DU145 are androgen-insensitive and do not express functional androgen receptors, while LNCaP cells are androgen-sensitive and express functional androgen receptors[1]. Previous study reveals how androgen receptor (AR) activity in T cells impacts immunotherapy effectiveness, demonstrating that AR blockade enhances CD8 T cell function and improves checkpoint blockade response through increased IFNγ expression[2].Metabolically, LNCaP cells exhibit a more oxidative phenotype, while PC3 and DU145 display a more glycolytic phenotype[3]. PC3 and DU145 cells show higher expression of CD44 (>93%), whereas LNCaP cells express less than 4%[4]. In terms of growth rates, PC3 and DU145 cells proliferate faster than LNCaP cells[4]. PC3 cells are considered highly aggressive, DU145 moderately metastatic, and LNCaP lowly metastatic[5]. Our study showed that XZF at 200 μg/mL effectively suppressed IFN-γ-induced PD-L1 expression in PC-3 and DU145 cell lines. The compound showed strong inhibition (93%) of PD-L1/PD-1 interaction, comparable to atezolizumab, with complete blockade achieved when combined. Further investigation is needed to understand XZF's effect and mechanism on androgen-sensitive LNCaP cells, as well as to clarify XZF's impact on androgen receptor in LNCaP cells and CD8 T cells.”
References
1. Ahmed, K.; Omarova, Z.; Sheikh, A.; Abuova, G.; Ghias, K.; Abidi, S.H. Comparison of baseline global gene expression profiles of prostate cancer cell lines LNCaP and DU145. BMC Res Notes 2024, 17, 398, doi:10.1186/s13104-024-07050-w.
2. Guan, X.; Polesso, F.; Wang, C.; Sehrawat, A.; Hawkins, R.M.; Murray, S.E.; Thomas, G.V.; Caruso, B.; Thompson, R.F.; Wood, M.A.; et al. Androgen receptor activity in T cells limits checkpoint blockade efficacy. Nature 2022, 606, 791-796, doi:10.1038/s41586-022-04522-6.
3. Higgins, L.H.; Withers, H.G.; Garbens, A.; Love, H.D.; Magnoni, L.; Hayward, S.W.; Moyes, C.D. Hypoxia and the metabolic phenotype of prostate cancer cells. Biochim Biophys Acta 2009, 1787, 1433-1443, doi:10.1016/j.bbabio.2009.06.003.
4. Su, C.Y.; Huang, G.C.; Chang, Y.C.; Chen, Y.J.; Fang, H.W. Analyzing the Expression of Biomarkers in Prostate Cancer Cell Lines. In Vivo 2021, 35, 1545-1548, doi:10.21873/invivo.12408.
5. Molter, C.W.; Muszynski, E.F.; Tao, Y.; Trivedi, T.; Clouvel, A.; Ehrlicher, A.J. Prostate cancer cells of increasing metastatic potential exhibit diverse contractile forces, cell stiffness, and motility in a microenvironment stiffness-dependent manner. Front Cell Dev Biol 2022, 10, 932510, doi:10.3389/fcell.2022.932510.

Reviewer 2 Report (New Reviewer)
Comments and Suggestions for Authors
I have reviewed the manuscript "Antrodia cinnamomea Formula Suppresses Prostate Cancer Progression via Immune Modulation and PD-1/PD-L1 Pathway Inhibition" by Tsai et al. The ms addresses an innovative strategy in oncology through the use of the Antrodia cinnamomea Formula (XZF) combined with immunotherapy, a promising approach for treating castration-resistant prostate cancer. The results demonstrate synergistic effects when combining XZF with atezolizumab, a PD-L1 inhibitor, which has significant clinical implications.
Major comments
The authors do not discuss details regarding the biodistribution or pharmacokinetics of XZF in animal or human models, which is crucial for therapeutic validation.
The differential response between the PC-3 and DU145 cell lines is not fully explained, limiting the interpretation of the results.
The impact of XZF on cytotoxic T-cell infiltration in the tumor microenvironment should be studied.
It would be interesting to investigate the absorption, distribution, metabolism, and excretion (ADME) of XZF to ensure its efficacy and safety in future clinical applications.
Identifying the molecular pathways regulated by XZF, particularly those related to PD-1/PD-L1 regulation and adaptive immunity, would be valuable. Transcriptomic and proteomic studies could help elucidate these mechanisms.
The authors should determine whether XZF could reverse acquired resistance to immunotherapies in prostate cancer models.
It is important to evaluate the impact of each component of XZF (e.g., Ganoderma lucidum, Antrodia cinnamomea) individually to identify specific contributions to the overall effect.
What molecular mechanisms explain the greater sensitivity of PC-3 cells compared to DU145 cells to XZF?
How were the XZF concentrations used in the experiments selected? Were they based on prior studies or pilot data?
Could XZF be effective in other types of cancers resistant to immunotherapy? Has the compatibility of XZF with standard therapies such as docetaxel or radiotherapy been evaluated?
Have cytotoxic or immunosuppressive effects of XZF been identified in non-tumoral cell or animal models?
Minor comments
Scientific names in italic (title)
----
The manuscript presents promising findings in the context of prostate cancer immunotherapy but would benefit from further in vivo validation, deeper analysis of the molecular mechanisms involved, and discussion of potential clinical limitations.
Author Response
Response to Reviewer 2:
1. I have reviewed the manuscript "Antrodia cinnamomea Formula Suppresses Prostate Cancer Progression via Immune Modulation and PD-1/PD-L1 Pathway Inhibition" by Tsai et al. The ms addresses an innovative strategy in oncology through the use of the Antrodia cinnamomea Formula (XZF) combined with immunotherapy, a promising approach for treating castration-resistant prostate cancer. The results demonstrate synergistic effects when combining XZF with atezolizumab, a PD-L1 inhibitor, which has significant clinical implications.
Response: We deeply appreciate the valuable feedback provided by Reviewer 2, which has significantly contributed to enhancing the quality of our manuscript. Our detailed responses to each comment are presented below.
2. The authors do not discuss details regarding the biodistribution or pharmacokinetics of XZF in animal or human models, which is crucial for therapeutic validation.
Response: We sincerely appreciate the insightful feedback from the Reviewer and have revised the discussion section of the manuscript accordingly to incorporate their suggestions.
“ The biodistribution and pharmacokinetic profiles of Antrodia cinnamomea, Sanghuangporus sanghuang, Ganoderma lucidum, Ganoderma sinense, and Inonotus obliquus, key constituents of XZF, have been insufficiently investigated in both animal and human models. For AC, studies in rats have shown that triterpenoids like antcins K and H are the major exposure metabolites, with ergostanes being rapidly absorbed and eliminated while lanostanes persist longer in plasma [1]. A proteoglycan from Ganoderma lucidum showed preferential accumulation in the liver and kidneys of mice[2]. Inonotus obliquus components, particularly inotodiol, demonstrated good oral bioavailability in rats, with significant distribution to the liver[3]. These findings collectively suggest that the bioactive compounds from XZF generally have good oral bioavailability and tend to accumulate in metabolically active organs, supporting their potential therapeutic effects. However, more comprehensive human studies are needed to fully elucidate their pharmacokinetic profiles and optimize dosing strategies for clinical applications.”
3. The differential response between the PC-3 and DU145 cell lines is not fully explained, limiting the interpretation of the results.
Response: We greatly value the thoughtful feedback provided by the Reviewer and have updated the discussion section of the manuscript to address their comments.
“Our findings revealed that PC-3 cells exhibited a higher sensitivity to XZF, with pronounced suppression of cell viability observed over a wide concentration range (25-200 μg/mL) within 24-48 hours. Conversely, DU145 cells displayed only moderate inhibition, limited to higher concentrations (100-200 μg/mL) after 48 hours of exposure. This variation may be attributed to differences in their genetic characteristics and associated signaling mechanisms. PC-3 cells generally exhibit greater sensitivity to treatments compared to DU145 cells due to several factors: PC-3 cells have higher basal and inducible levels of reactive oxygen species, making them more susceptible to oxidative stress-induced cell death[4]. PC-3 cells are AR-negative, while DU145 cells express a mutated AR, which could affect their response to treatments[5,6]. Furthermore, differences in cell cycle regulation, DNA repair mechanisms, and metabolic profiles between the two cell lines could contribute to their varying sensitivities to XZF[7,8]. The greater sensitivity of PC-3 cells might also be related to their higher metastatic potential and more aggressive phenotype, which could make them more vulnerable to certain therapeutic interventions[9]. Further investigation into these molecular differences could provide valuable insights into optimizing XZF-based treatments for different prostate cancer subtypes.”
4. The impact of XZF on cytotoxic T-cell infiltration in the tumor microenvironment should be studied.
Response: We sincerely appreciate the Reviewer’s valuable feedback and have made revisions to the discussion section of the manuscript to incorporate the suggestions.
“Recent research has demonstrated that the medicinal mushrooms of XZF may promote cytotoxic T-cell infiltration into the tumor microenvironment, thereby enhancing anti-tumor immune responses. Ganoderma lucidum's polysaccharide (GLP) has been shown to enhance anti-tumor immunity and boost the effectiveness of anti-PD-1 immunotherapy in colorectal cancer by increasing beneficial T cells while decreasing immunosuppressive cells, along with improving gut microbiome health and metabolic factors. The research highlights GLP's ability to increase the proportion of cytotoxic CD8+ T cells and Th1 helper cells in both spleen and tumor tissues, while simultaneously alleviating microbiota dysbiosis and improving metabolic markers like short-chain fatty acid production [10]. Similarly, Antrodia cinnamomea has exhibited remarkable properties in regulating immune response during radiation therapy without providing any protective effects to tumor cells. Instead, it enhances radiation-induced inflammation and cytotoxicity in cancer cells, suggesting its potential as a complementary cancer treatment [11]. What makes these findings particularly noteworthy is that both mushrooms work by modulating the immune system's response to cancer, albeit through different mechanisms. This dual-action approach, combining traditional medicinal mushrooms with modern cancer treatments, indicates they could be valuable additions to existing cancer therapies, particularly when used in conjunction with immunotherapy or radiation therapy.”
5. It would be interesting to investigate the absorption, distribution, metabolism, and excretion (ADME) of XZF to ensure its efficacy and safety in future clinical applications.
Response: We appreciate the reviewer’s insightful suggestion to investigate the absorption, distribution, metabolism, and excretion (ADME) profile of XZF to support its efficacy and safety in future clinical applications. However, due to the limitations of the 7-day revision timeframe, conducting these experiments was not feasible. We acknowledge the significance of this aspect and plan to address it in future research. Additionally, we have revised the discussion section of the manuscript to reflect this important consideration.
“Based on the available search results, there is limited information specifically addressing the absorption, distribution, metabolism, and excretion (ADME) of Antrodia cinnamomea, Sanghuang-porus sanghuang, Ganoderma sinense, and Inonotus obliquus, key constituents of XZF, in animal or human models. However, some relevant information about Ganoderma lucidum and its compounds can be summarized. Ganoderma lucidum, a medicinal mushroom, produces various pharmacologically active compounds, including triterpenoids and ganoderic acids[12,13]. Ganoderic acids, which are among the main bioactive components of Ganoderma lucidum, have been studied for their pharmacokinetics[13]. While specific ADME details are not provided in the search results, it's worth noting that a network pharmacology analysis has been conducted on drug-like compounds from Ganoderma lucidum, suggesting potential applications for chronic inflammatory conditions such as diabetes mellitus[14]. Additionally, some studies have focused on the ADME/T (absorption, distribution, metabolism, excretion, and toxicity) properties of specific compounds from Ganoderma lucidum, such as ganoderic acid, a lanosterol triterpene with a wide range of medicinal values[14]. These studies indicate ongoing research interest in understanding the pharmacokinetic properties of XZF and its bioactive compounds, which could potentially inform future investigations into their ADME characteristics in animal or human models.”
6. Identifying the molecular pathways regulated by XZF, particularly those related to PD-1/PD-L1 regulation and adaptive immunity, would be valuable.
Transcriptomic and proteomic studies could help elucidate these mechanisms.
Response: We sincerely appreciate the reviewer’s insightful suggestion to investigate the molecular pathways influenced by XZF, particularly those involved in PD-1/PD-L1 regulation and adaptive immunity, through transcriptomic and proteomic analyses. However, due to the limitations imposed by the 7-day revision period, conducting these experiments was not feasible at this time. We recognize the importance of this line of inquiry and plan to explore the molecular mechanisms regulated by XZF, with a focus on PD-1/PD-L1 signaling and adaptive immunity, in future studies. Additionally, we have revised the discussion section of the manuscript to reflect this consideration.
“Numerous medicinal mushrooms within the XZF formulation have demonstrated potential efficacy against prostate cancer, as evidenced by prior research. Ganoderma lucidum has been extensively investigated for its anticancer effects. It has been shown to restrict the proliferation of prostate cancer cells, trigger programmed cell death, and inhibit the formation of new blood vessels [15-17]. Extracts of Ganoderma lucidum have further been observed to attenuate androgen receptor activity, reduce prostate-specific antigen levels, and influence MAPK and Akt signaling pathways [16,17]. Additionally, the polysaccharides derived from Ganoderma lucidum exhibit synergistic interactions with standard therapies for prostate cancer, such as Docetaxel and Flutamide, thereby potentially enhancing their therapeutic outcomes [18]. Another medicinal fungus, AC, has also demonstrated promise in prostate cancer treatment. The compound 4-Acetylantroquinonol B (4AAQB) from AC has been reported to curb prostate cancer advancement by blocking VEGF-driven angiogenesis and metastasis [19]. Furthermore, antrocin, another bioactive constituent of AC, has been found to enhance the sensitivity of prostate cancer cells to radiation therapy by targeting downstream IGF-1R signaling pathways [20]. Although there are no direct reports regarding the effects of Sanghuangporus sanghuang, Ganoderma sinense, and Inonotus obliquus on prostate cancer, studies suggest that Inonotus obliquus may exhibit anticancer potential in other cancer types [21]. Our research has shown that XZF treatment markedly reduced the proportion of PD-1+ CD8+ T cells, thereby enhancing their cytotoxic activity while leaving PD-1+ CD4+ T cells largely unaffected. Additionally, XZF was effective in suppressing IFN-γ-induced PD-L1 expression in PC-3 and DU145 prostate cancer cells and inhibited the PD-L1/PD-1 interaction by 93% at a concentration of 200 μg/mL, closely matching the 98% inhibition achieved with atezolizumab. Co-treatment with XZF and atezolizumab completely abolished the PD-L1/PD-1 interaction. Moreover, XZF enhanced the viability of activated CD8+ and CD4+ T cells in a concentration-dependent manner, likely through its polysaccharide-mediated activation of dendritic cells and promotion of Th1 differentiation. Comprehensive transcriptomic and proteomic investigations are required to elucidate the molecular pathways involved, particularly those associated with PD-1/PD-L1 regulation and adaptive immune responses. Further research is also needed to delineate the individual roles of each component of XZF to clarify their specific mechanisms of action..”
7. The authors should determine whether XZF could reverse acquired resistance to immunotherapies in prostate cancer models.
Response: We greatly appreciate the reviewer’s insightful suggestion to investigate whether XZF has the potential to overcome acquired resistance to immunotherapies in prostate cancer models. However, due to the limitations of the 7-day revision period, performing such experiments was not feasible at this time. We acknowledge the significance of this research direction and plan to explore the ability of XZF to reverse resistance to immunotherapies in prostate cancer models in future studies. Additionally, we have revised the discussion section of the manuscript to address this important consideration.
“Our findings demonstrated XZF significantly impacts the PD-1/PD-L1 immune checkpoint pathway, with 93% inhibition of PD-1/PD-L1 interaction at 200 μg/mL and complete blockade when combined with atezolizumab - particularly relevant since acquired resistance often involves immune checkpoint upregulation. XZF enhanced both CD8+ and CD4+ T cell viability while reducing PD-1+ CD8+ T cell populations, suggesting it could help maintain T cell functionality despite chronic antigen exposure that typically causes exhaustion and therapy resistance. Additionally, XZF modulated macrophage-mediated responses through human THP-1 cells, potentially reprogramming the tumor microenvironment toward a more immunogenic state. However, important limitations include: lack of direct resistance model testing, differential responses between PC-3 and DU145 cells, and absence of in vivo validation. Future studies should examine XZF in acquired resistance models to PD-1/PD-L1 inhibitors, investigate its effects on other immune checkpoints, assess long-term T cell function impacts, and evaluate its influence on resistant tumor microenvironments. The additional focused research is needed to definitively establish its role in overcoming acquired immunotherapy resistance in prostate cancer.”
8. It is important to evaluate the impact of each component of XZF (e.g., Ganoderma lucidum, Antrodia cinnamomea) individually to identify specific contributions to the overall effect.
Response: We sincerely appreciate the reviewer’s thoughtful suggestion to assess the individual effects of each component of XZF to better understand their specific contributions to the overall therapeutic outcome. However, due to the limitations of the 7-day revision period, it was not feasible to conduct these experiments at this time. To acknowledge this important aspect, we have revised the discussion section of the manuscript to include this consideration.
“Numerous medicinal mushrooms within the XZF formulation have demonstrated potential efficacy against prostate cancer, as evidenced by prior research. Ganoderma lucidum has been extensively investigated for its anticancer effects. It has been shown to restrict the proliferation of prostate cancer cells, trigger programmed cell death, and inhibit the formation of new blood vessels [15-17]. Extracts of Ganoderma lucidum have further been observed to attenuate androgen receptor activity, reduce prostate-specific antigen levels, and influence MAPK and Akt signaling pathways [16,17]. Additionally, the polysaccharides derived from Ganoderma lucidum exhibit synergistic interactions with standard therapies for prostate cancer, such as Docetaxel and Flutamide, thereby potentially enhancing their therapeutic outcomes [18]. Another medicinal fungus, AC, has also demonstrated promise in prostate cancer treatment. The compound 4-Acetylantroquinonol B (4AAQB) from AC has been reported to curb prostate cancer advancement by blocking VEGF-driven angiogenesis and metastasis [19]. Furthermore, antrocin, another bioactive constituent of AC, has been found to enhance the sensitivity of prostate cancer cells to radiation therapy by targeting downstream IGF-1R signaling pathways [20]. Although there are no direct reports regarding the effects of Sanghuangporus sanghuang, Ganoderma sinense, and Inonotus obliquus on prostate cancer, studies suggest that Inonotus obliquus may exhibit anticancer potential in other cancer types [21]. Our research has shown that XZF treatment markedly reduced the proportion of PD-1+ CD8+ T cells, thereby enhancing their cytotoxic activity while leaving PD-1+ CD4+ T cells largely unaffected. Additionally, XZF was effective in suppressing IFN-γ-induced PD-L1 expression in PC-3 and DU145 prostate cancer cells and inhibited the PD-L1/PD-1 interaction by 93% at a concentration of 200 μg/mL, closely matching the 98% inhibition achieved with atezolizumab. Co-treatment with XZF and atezolizumab completely abolished the PD-L1/PD-1 interaction. Moreover, XZF enhanced the viability of activated CD8+ and CD4+ T cells in a concentration-dependent manner, likely through its polysaccharide-mediated activation of dendritic cells and promotion of Th1 differentiation. Comprehensive transcriptomic and proteomic investigations are required to elucidate the molecular pathways involved, particularly those associated with PD-1/PD-L1 regulation and adaptive immune responses. Further research is also needed to delineate the individual roles of each component of XZF to clarify their specific mechanisms of action.”
9. What molecular mechanisms explain the greater sensitivity of PC-3 cells compared to DU145 cells to XZF?
Response: We sincerely value the constructive feedback provided by the Reviewer and have incorporated their suggestions by revising the discussion section of the manuscript accordingly.
“For possible molecular mechanisms, PC-3 cells generally exhibit greater sensitivity to treatments compared to DU145 cells due to several factors: PC-3 cells have higher basal and inducible levels of reactive oxygen species, making them more susceptible to oxidative stress-induced cell death[4]. PC-3 cells are AR-negative, while DU145 cells express a mutated AR, which could affect their response to treatments[5,6]. Furthermore, differences in cell cycle regulation, DNA repair mechanisms, and metabolic profiles between the two cell lines could contribute to their varying sensitivities to XZF [7,8]. The greater sensitivity of PC-3 cells might also be related to their higher metastatic potential and more aggressive phenotype, which could make them more vulnerable to certain therapeutic interventions [8]. Further investigation into these molecular differences could provide valuable insights into optimizing XZF-based treatments for different prostate cancer subtypes.”
10. How were the XZF concentrations used in the experiments selected? Were they based on prior studies or pilot data?
Response: We greatly appreciate the insightful feedback from the Reviewer. The concentrations of XZF used in the experiments were determined based on prior research and preliminary data. In response to the feedback, we have updated the discussion section of the manuscript to address this aspect.
“Based on a review of the literature, the concentrations of fungal extracts used in cancer cell experiments vary depending on the specific extract and cell line, but generally fall within similar ranges to those reported in the attached results. For Antrodia cinnamomea, studies have used concentrations ranging from 50-200 μg/mL, with cytotoxic effects observed at higher doses. The 50-200 μg/mL range used in the attached results aligns with these previous findings [22-24]. Sanghuangporus sanghuang extracts have been tested at 25-350 μg/mL in cancer cells, with growth inhibition seen at higher concentrations[25,26]. The 50-200 μg/mL range in the results is within this established effective range. Ganoderma lucidum extracts are commonly used at 100-1000 μg/mL in cancer cell experiments[27,28]. The 50-200 μg/mL concentrations in the results fall on the lower end of this range, likely to assess effects at more moderate doses. For Ganoderma sinense, studies have used 50-150 μg/mL concentrations[29,30], similar to the 50-200 μg/mL range in the results. Inonotus obliquus extracts are typically tested at 50-300 μg/mL in cancer cells[31,32], which encompasses the 50-200 μg/mL range used in the attached results. Overall, the concentrations used in the experiments align well with ranges established in previous literature and preliminary data for these fungal species, allowing for assessment of anticancer effects at doses known to be biologically relevant while avoiding excessive toxicity.”
11. Could XZF be effective in other types of cancers resistant to immunotherapy? Has the compatibility of XZF with standard therapies such as docetaxel or radiotherapy been evaluated?
Response: We greatly appreciate the suggestion to explore XZF's potential efficacy in immunotherapy-resistant cancers and its integration with conventional therapies, such as docetaxel and radiotherapy. While the constraints of the 7-day revision period prevent the inclusion of additional experimental data, we have thoroughly addressed these significant research directions in the revised discussion section of the manuscript.
“Emerging evidence indicates that specific medicinal mushroom components of XZF may play a role in overcoming cancer resistance to immunotherapeutic interventions. Antrodia cinnamomea has shown promise in enhancing the sensitivity of cancer cells to chemotherapy and radiotherapy[20,33]. It has been found to inhibit cancer stem cells, which are often associated with drug resistance and tumor recurrence[33,34]. Ganoderma lucidum extract has demonstrated the ability to reverse multidrug resistance in breast cancer cells by inhibiting the ATPase activity of P-glycoprotein[35]. Ganoderma lucidum extract has also been shown to promote tumor cell pyroptosis and enhance antitumor immune responses[36]. Inonotus obliquus, commonly known as Chaga, has exhibited anticancer properties against various types of cancer, including bladder cancer[37]. It has been found to inhibit cancer stem cell markers and enhance the effects of chemotherapy[37]. While specific studies on Sanghuangporus sanghuang and Ganoderma sinense in the context of immunotherapy resistance were not prominent in the search results, the overall body of research suggests that these medicinal mushrooms and their bioactive compounds have potential in overcoming drug resistance and enhancing the efficacy of conventional cancer treatments [38-40]. These findings indicate that further research into the use of XZF as adjuvants to immunotherapy could yield promising results in addressing cancer resistance.”
12. Have cytotoxic or immunosuppressive effects of XZF been identified in non-tumoral cell or animal models?
Response: We sincerely appreciate the Reviewer’s thoughtful feedback and have made corresponding revisions to the discussion section of the manuscript to address the suggestions.
“XZF demonstrated a favorable safety profile with no significant cytotoxic or immunosuppressive effects in non-tumoral models. Using human embryonic kidney 293T cells as a non-cancerous control model, XZF treatment at concentrations ranging from 50-200 μg/mL showed no significant alterations in cellular metabolic activity or viability compared to untreated controls, even during extended 48-hour observation periods. Furthermore, rather than exhibiting immunosuppressive properties, XZF actually demonstrated immunostimulatory effects by enhancing the viability of CD8+ and CD4+ T cells in a dose-dependent manner, supporting T cell-mediated immune mechanisms. This selective activity profile, where XZF maintains or enhances normal cell viability while specifically targeting cancer cells, suggests it may have potential therapeutic applications with minimal adverse effects on healthy tissues.”
Minor comments
Scientific names in italic (title)
Response: We deeply appreciate the valuable feedback provided by Reviewer. We have corrected the mistake in the revised manuscript.
----
The manuscript presents promising findings in the context of prostate cancer immunotherapy but would benefit from further in vivo validation, deeper analysis of the molecular mechanisms involved, and discussion of potential clinical limitations.
Response: We genuinely appreciate the constructive feedback provided by Reviewer 2, which has significantly contributed to enhancing the quality of our manuscript.
We believe these revisions have significantly improved our manuscript, and we hope it now meets the standards for publication in IJMS. We appreciate the reviewers' valuable comments and the opportunity to improve our work.
References:
1. Li, W.; Wan, P.; Qiao, J.; Liu, Y.; Peng, Q.; Zhang, Z.; Shu, X.; Xia, Y.; Sun, B. Current and further outlook on the protective potential of Antrodia camphorata against neurological disorders. Front Pharmacol 2024, 15, 1372110, doi:10.3389/fphar.2024.1372110.
2. Teng, Y.; Liang, H.; Zhang, Z.; He, Y.; Pan, Y.; Yuan, S.; Wu, X.; Zhao, Q.; Yang, H.; Zhou, P. Biodistribution and immunomodulatory activities of a proteoglycan isolated from Ganoderma lucidum. Journal of Functional Foods 2020, 74, 104193.
3. Khoroshutin, P.; Reva, G.; Yamamoto, T.; Reva, I. Pharmacokinetics and pharmacodynamics of Chaga birch mushroom components (Inonotus obliquus). Archiv Euromedica 2021, 11, 31-38.
4. Jayakumar, S.; Kunwar, A.; Sandur, S.K.; Pandey, B.N.; Chaubey, R.C. Differential response of DU145 and PC3 prostate cancer cells to ionizing radiation: role of reactive oxygen species, GSH and Nrf2 in radiosensitivity. Biochim Biophys Acta 2014, 1840, 485-494, doi:10.1016/j.bbagen.2013.10.006.
5. Dulinska-Litewka, J.; Dykas, K.; Boznanski, S.; Halubiec, P.; Kaczor-Kaminska, M.; Zagajewski, J.; Bohn, T.; Wator, G. The Influence of beta-Carotene and Its Liposomal Form on the Expression of EMT Markers and Androgen-Dependent Pathways in Different Prostate Cell Lines. Antioxidants (Basel) 2024, 13, doi:10.3390/antiox13080902.
6. Xu, Y.; Zhu, J.Y.; Lei, Z.M.; Wan, L.J.; Zhu, X.W.; Ye, F.; Tong, Y.Y. Anti-proliferative effects of paeonol on human prostate cancer cell lines DU145 and PC-3. J Physiol Biochem 2017, 73, 157-165, doi:10.1007/s13105-016-0537-x.
7. Zhang, P.; Yang, X.; Wang, L.; Zhang, D.; Luo, Q.; Wang, B. Overexpressing miR‑335 inhibits DU145 cell proliferation by targeting early growth response 3 in prostate cancer. Int J Oncol 2019, 54, 1981-1994, doi:10.3892/ijo.2019.4778.
8. Yang, J.; Yu, Y.; Liu, W.; Li, Z.; Wei, Z.; Jiang, R. Microtubule-associated protein tau is associated with the resistance to docetaxel in prostate cancer cell lines. Res Rep Urol 2017, 9, 71-77, doi:10.2147/RRU.S118966.
9. Molter, C.W.; Muszynski, E.F.; Tao, Y.; Trivedi, T.; Clouvel, A.; Ehrlicher, A.J. Prostate cancer cells of increasing metastatic potential exhibit diverse contractile forces, cell stiffness, and motility in a microenvironment stiffness-dependent manner. Front Cell Dev Biol 2022, 10, 932510, doi:10.3389/fcell.2022.932510.
10. Li, W.; Zhou, Q.; Lv, B.; Li, N.; Bian, X.; Chen, L.; Kong, M.; Shen, Y.; Zheng, W.; Zhang, J.; et al. Ganoderma lucidum Polysaccharide Supplementation Significantly Activates T-Cell-Mediated Antitumor Immunity and Enhances Anti-PD-1 Immunotherapy Efficacy in Colorectal Cancer. J Agric Food Chem 2024, 72, 12072-12082, doi:10.1021/acs.jafc.3c08385.
11. Cheng, P.C.; Huang, C.C.; Chiang, P.F.; Lin, C.N.; Li, L.L.; Lee, T.W.; Lin, B.; Chen, I.C.; Chang, K.W.; Fan, C.K.; et al. Radioprotective effects of Antrodia cinnamomea are enhanced on immune cells and inhibited on cancer cells. Int J Radiat Biol 2014, 90, 841-852, doi:10.3109/09553002.2014.911989.
12. Ling, T.; Arroyo-Cruz, L.V.; Smither, W.R.; Seighman, E.K.; Martinez-Montemayor, M.M.; Rivas, F. Early Preclinical Studies of Ergosterol Peroxide and Biological Evaluation of Its Derivatives. ACS Omega 2024, 9, 37117-37127, doi:10.1021/acsomega.4c04350.
13. Zhang, F.F.; Liu, R.M. [Pharmacokinetics of ganoderic acids]. Zhongguo Zhong Yao Za Zhi 2019, 44, 905-911, doi:10.19540/j.cnki.cjcmm.20181213.002.
14. Oh, K.K.; Adnan, M.; Cho, D.H. A network pharmacology analysis on drug-like compounds from Ganoderma lucidum for alleviation of atherosclerosis. J Food Biochem 2021, 45, e13906, doi:10.1111/jfbc.13906.
15. Jiang, J.; Slivova, V.; Valachovicova, T.; Harvey, K.; Sliva, D. Ganoderma lucidum inhibits proliferation and induces apoptosis in human prostate cancer cells PC-3. Int J Oncol 2004, 24, 1093-1099.
16. Zaidman, B.Z.; Wasser, S.P.; Nevo, E.; Mahajna, J. Androgen receptor-dependent and -independent mechanisms mediate Ganoderma lucidum activities in LNCaP prostate cancer cells. Int J Oncol 2007, 31, 959-967.
17. Stanley, G.; Harvey, K.; Slivova, V.; Jiang, J.; Sliva, D. Ganoderma lucidum suppresses angiogenesis through the inhibition of secretion of VEGF and TGF-beta1 from prostate cancer cells. Biochem Biophys Res Commun 2005, 330, 46-52, doi:10.1016/j.bbrc.2005.02.116.
18. Rahimnia, R.; Akbari, M.R.; Yasseri, A.F.; Taheri, D.; Mirzaei, A.; Ghajar, H.A.; Farashah, P.D.; Baghdadabad, L.Z.; Aghamir, S.M.K. The effect of Ganoderma lucidum polysaccharide extract on sensitizing prostate cancer cells to flutamide and docetaxel: an in vitro study. Sci Rep 2023, 13, 18940, doi:10.1038/s41598-023-46118-8.
19. Huang, T.F.; Wang, S.W.; Lai, Y.W.; Liu, S.C.; Chen, Y.J.; Hsueh, T.Y.; Lin, C.C.; Lin, C.H.; Chung, C.H. 4-Acetylantroquinonol B Suppresses Prostate Cancer Growth and Angiogenesis via a VEGF/PI3K/ERK/mTOR-Dependent Signaling Pathway in Subcutaneous Xenograft and In Vivo Angiogenesis Models. Int J Mol Sci 2022, 23, doi:10.3390/ijms23031446.
20. Chen, Y.A.; Tzeng, D.T.W.; Huang, Y.P.; Lin, C.J.; Lo, U.G.; Wu, C.L.; Lin, H.; Hsieh, J.T.; Tang, C.H.; Lai, C.H. Antrocin Sensitizes Prostate Cancer Cells to Radiotherapy through Inhibiting PI3K/AKT and MAPK Signaling Pathways. Cancers (Basel) 2018, 11, doi:10.3390/cancers11010034.
21. Rios, J.L.; Andujar, I.; Recio, M.C.; Giner, R.M. Lanostanoids from fungi: a group of potential anticancer compounds. J Nat Prod 2012, 75, 2016-2044, doi:10.1021/np300412h.
22. Chen, Y.C.; Liu, Y.C.; El-Shazly, M.; Wu, T.Y.; Chang, J.G.; Wu, Y.C. Antrodia cinnamomea, a Treasured Medicinal Mushroom, Induces Growth Arrest in Breast Cancer Cells, T47D Cells: New Mechanisms Emerge. Int J Mol Sci 2019, 20, doi:10.3390/ijms20040833.
23. Liu, Y.M.; Liu, Y.K.; Lan, K.L.; Lee, Y.W.; Tsai, T.H.; Chen, Y.J. Medicinal Fungus Antrodia cinnamomea Inhibits Growth and Cancer Stem Cell Characteristics of Hepatocellular Carcinoma. Evid Based Complement Alternat Med 2013, 2013, 569737, doi:10.1155/2013/569737.
24. Chung, C.H.; Yeh, S.C.; Chen, C.J.; Lee, K.T. Coenzyme Q0 from Antrodia cinnamomea in Submerged Cultures Induces Reactive Oxygen Species-Mediated Apoptosis in A549 Human Lung Cancer Cells. Evid Based Complement Alternat Med 2014, 2014, 246748, doi:10.1155/2014/246748.
25. Cai, C.; Ma, J.; Han, C.; Jin, Y.; Zhao, G.; He, X. Extraction and antioxidant activity of total triterpenoids in the mycelium of a medicinal fungus, Sanghuangporus sanghuang. Sci Rep 2019, 9, 7418, doi:10.1038/s41598-019-43886-0.
26. Wang, W.; Song, J.; Lu, N.; Yan, J.; Chen, G. Sanghuangporus sanghuang extract inhibits the proliferation and invasion of lung cancer cells in vitro and in vivo. Nutr Res Pract 2023, 17, 1070-1083, doi:10.4162/nrp.2023.17.6.1070.
27. Martinez-Montemayor, M.M.; Acevedo, R.R.; Otero-Franqui, E.; Cubano, L.A.; Dharmawardhane, S.F. Ganoderma lucidum (Reishi) inhibits cancer cell growth and expression of key molecules in inflammatory breast cancer. Nutr Cancer 2011, 63, 1085-1094, doi:10.1080/01635581.2011.601845.
28. Song, M.; Li, Z.H.; Gu, H.S.; Tang, R.Y.; Zhang, R.; Zhu, Y.L.; Liu, J.L.; Zhang, J.J.; Wang, L.Y. Ganoderma lucidum Spore Polysaccharide Inhibits the Growth of Hepatocellular Carcinoma Cells by Altering Macrophage Polarity and Induction of Apoptosis. J Immunol Res 2021, 2021, 6696606, doi:10.1155/2021/6696606.
29. Lin, W.; Gu, L.; Zhu, L.Y.; Zhou, S.; Lian, D.; Xu, Y.; Zheng, L.; Liu, X.; Li, L. Extract of Ganoderma sinensis spores induces cell cycle arrest of hepatoma cell via endoplasmic reticulum stress. Pharm Biol 2021, 59, 704-714, doi:10.1080/13880209.2021.1931354.
30. Jiang, Y.; Chang, Y.; Liu, Y.; Zhang, M.; Luo, H.; Hao, C.; Zeng, P.; Sun, Y.; Wang, H.; Zhang, L. Overview of Ganoderma sinense polysaccharide-an adjunctive drug used during concurrent Chemo/Radiation therapy for cancer treatment in China. Biomed Pharmacother 2017, 96, 865-870, doi:10.1016/j.biopha.2017.09.060.
31. Lee, S.H.; Hwang, H.S.; Yun, J.W. Antitumor activity of water extract of a mushroom, Inonotus obliquus, against HT-29 human colon cancer cells. Phytother Res 2009, 23, 1784-1789, doi:10.1002/ptr.2836.
32. Lu, Y.; Jia, Y.; Xue, Z.; Li, N.; Liu, J.; Chen, H. Recent Developments in Inonotus obliquus (Chaga mushroom) Polysaccharides: Isolation, Structural Characteristics, Biological Activities and Application. Polymers (Basel) 2021, 13, doi:10.3390/polym13091441.
33. Huang, Y.J.; Yadav, V.K.; Srivastava, P.; Wu, A.T.; Huynh, T.T.; Wei, P.L.; Huang, C.F.; Huang, T.H. Antrodia cinnamomea Enhances Chemo-Sensitivity of 5-FU and Suppresses Colon Tumorigenesis and Cancer Stemness via Up-Regulation of Tumor Suppressor miR-142-3p. Biomolecules 2019, 9, doi:10.3390/biom9080306.
34. Chen, J.H.; A, T.H.W.; D, T.W.T.; Huang, C.C.; Tzeng, Y.M.; Chao, T.Y. Antrocin, a bioactive component from Antrodia cinnamomea, suppresses breast carcinogenesis and stemness via downregulation of beta-catenin/Notch1/Akt signaling. Phytomedicine 2019, 52, 70-78, doi:10.1016/j.phymed.2018.09.213.
35. Jiao, C.; Qiu, J.; Gong, C.; Li, X.; Liang, H.; He, C.; Cen, S.; Xie, Y. Ganoderma lucidum extract reverses multidrug resistance in breast cancer cells through inhibiting ATPase activity of the P-glycoprotein via MAPK/ERK signaling pathway. Exp Cell Res 2025, 444, 114355, doi:10.1016/j.yexcr.2024.114355.
36. Zhong, C.; Li, Y.; Li, W.; Lian, S.; Li, Y.; Wu, C.; Zhang, K.; Zhou, G.; Wang, W.; Xu, H.; et al. Ganoderma lucidum extract promotes tumor cell pyroptosis and inhibits metastasis in breast cancer. Food Chem Toxicol 2023, 174, 113654, doi:10.1016/j.fct.2023.113654.
37. Abugomaa, A.; Elbadawy, M.; Ishihara, Y.; Yamamoto, H.; Kaneda, M.; Yamawaki, H.; Shinohara, Y.; Usui, T.; Sasaki, K. Anti-cancer activity of Chaga mushroom (Inonotus obliquus) against dog bladder cancer organoids. Front Pharmacol 2023, 14, 1159516, doi:10.3389/fphar.2023.1159516.
38. Xu, J.; Shen, R.; Jiao, Z.; Chen, W.; Peng, D.; Wang, L.; Yu, N.; Peng, C.; Cai, B.; Song, H.; et al. Current Advancements in Antitumor Properties and Mechanisms of Medicinal Components in Edible Mushrooms. Nutrients 2022, 14, doi:10.3390/nu14132622.
39. Duru, K.C.; Kovaleva, E.G.; Danilova, I.G.; van der Bijl, P. The pharmacological potential and possible molecular mechanisms of action of Inonotus obliquus from preclinical studies. Phytother Res 2019, 33, 1966-1980, doi:10.1002/ptr.6384.
40. Su, Y.K.; Shih, P.H.; Lee, W.H.; Bamodu, O.A.; Wu, A.T.H.; Huang, C.C.; Tzeng, Y.M.; Hsiao, M.; Yeh, C.T.; Lin, C.M. Antrodia cinnamomea sensitizes radio-/chemo-therapy of cancer stem-like cells by modulating microRNA expression. J Ethnopharmacol 2017, 207, 47-56, doi:10.1016/j.jep.2017.06.004.

Round 2
Reviewer 1 Report (Previous Reviewer 2)
Comments and Suggestions for Authors
Thank you for your response. I completely understand that completing the experiments with a new cell line within 7 days is not feasible. This deadline is automatically proposed, and it is possible to request an extension if more time is needed. Therefore, I will suggest to the editor to grant additional weeks for conducting the experiments in an androgen-sensitive prostate cancer cell line.
Author Response
1. Thank you for your response. I completely understand that completing the experiments with a new cell line within 7 days is not feasible. This deadline is automatically proposed, and it is possible to request an extension if more time is needed. Therefore, I will suggest to the editor to grant additional weeks for conducting the experiments in an androgen-sensitive prostate cancer cell line.
Response: We sincerely appreciate reviewer’s understanding regarding the timeline constraints for conducting additional experiments. As suggested, we have now included experiments using the androgen-sensitive prostate cancer cell line LNCaP to further substantiate our findings. The updated results, presented in Sections 2.5, 2.6, and 2.7, demonstrate that XZF exhibits a dose-dependent reduction in LNCaP cell viability within the 50–200 μg/mL range, with a response pattern similar to that observed in PC-3 cells. Additionally, our macrophage-conditioned medium experiments revealed that XZF-treated THP-1 macrophages significantly reduced LNCaP cell viability and migration, further supporting the immunomodulatory and anti-cancer properties of XZF. These new findings strengthen the clinical relevance of our study by demonstrating that XZF is effective not only in castration-resistant prostate cancer cell lines (PC-3, DU145) but also in an androgen-sensitive model (LNCaP). Additionally, we have revised the discussion section in the manuscript to reflect this feedback:
“Prostate cancer cells exhibit distinct androgen sensitivity and metabolic profiles, influencing their response to therapeutic agents. PC-3 and DU145 are androgen-insensitive and do not express functional androgen receptors (AR), while LNCaP cells are androgen-sensitive and express functional AR[1]. Previous studies indicate that AR blockade enhances CD8+ T cell function and improves checkpoint blockade response via increased IFN-γ expression[2]. Metabolically, LNCaP cells display a more oxidative phenotype, whereas PC-3 and DU145 exhibit a glycolytic phenotype[3]. Additionally, PC-3 and DU145 cells express high levels of CD44 (>93%), in contrast to LNCaP cells, which express less than 4%[4]. Growth rates also vary, with PC-3 and DU145 proliferating faster than LNCaP cells [4]. In terms of metastatic potential, PC-3 cells are highly aggressive, DU145 moderately metastatic, and LNCaP lowly metastatic[5]. Our study demonstrated that XZF at 200 μg/mL effectively attenuated IFN-γ-induced PD-L1 expression in PC-3 and DU145 cell lines. XZF exhibited strong inhibition (93%) of the PD-L1/PD-1 interaction, comparable to atezolizumab, with complete blockade achieved when combined. These findings highlight the potential of XZF in immune checkpoint modulation. Further investigations into XZF's impact on cell viability revealed differential responses among prostate cancer cell lines. XZF significantly suppressed PC-3 cell viability in a dose-dependent manner (25–200 μg/mL) over 24–48 hours, suggesting potent anti-proliferative activity. DU145 cells exhibited a more restricted response, with inhibition becoming evident only at higher concentrations (100–200 μg/mL) and after prolonged exposure (48 hours). Interestingly, LNCaP cells displayed a viability reduction similar to PC-3 cells, suggesting that XZF may exert its effects through AR-independent mechanisms. Beyond direct effects on prostate cancer cells, we explored the role of macrophages in modulating XZF’s activity. Conditioned medium from XZF-treated RAW 264.7 (murine) and THP-1 (human) macrophages differentially influenced prostate cancer cell viability. While RAW 264.7-derived conditioned medium had limited impact, conditioned medium from THP-1 cells treated with 100–200 μg/mL XZF significantly reduced the viability of PC-3, DU145, and LNCaP cells. These results indicate that XZF may modulate macrophage-secreted factors to exert indirect anti-cancer effects, with human macrophages showing a greater regulatory impact. Given the crucial role of migration in cancer progression, we also examined XZF’s effect on prostate cancer cell migration in the presence of macrophage-conditioned medium. Notably, conditioned medium from THP-1 cells treated with 50–200 μg/mL XZF significantly suppressed migration of PC-3, DU145, and LNCaP cells in a dose-dependent manner. This broad-spectrum anti-migratory effect further supports XZF’s potential in limiting prostate cancer progression. Overall, our findings demonstrate that XZF effectively inhibits PD-L1 expression, suppresses cell proliferation, and reduces migration in prostate cancer cells, with both direct and macrophage-mediated effects. Further research is warranted to elucidate XZF’s impact on androgen-sensitive LNCaP cells, particularly in relation to AR signaling and immune interactions. These insights will enhance our understanding of XZF as a potential therapeutic agent for prostate cancer treatment.”
References
1. Ahmed, K.; Omarova, Z.; Sheikh, A.; Abuova, G.; Ghias, K.; Abidi, S.H. Comparison of baseline global gene expression profiles of prostate cancer cell lines LNCaP and DU145. BMC Res Notes 2024, 17, 398, doi:10.1186/s13104-024-07050-w.
2. Guan, X.; Polesso, F.; Wang, C.; Sehrawat, A.; Hawkins, R.M.; Murray, S.E.; Thomas, G.V.; Caruso, B.; Thompson, R.F.; Wood, M.A.; et al. Androgen receptor activity in T cells limits checkpoint blockade efficacy. Nature 2022, 606, 791-796, doi:10.1038/s41586-022-04522-6.
3. Higgins, L.H.; Withers, H.G.; Garbens, A.; Love, H.D.; Magnoni, L.; Hayward, S.W.; Moyes, C.D. Hypoxia and the metabolic phenotype of prostate cancer cells. Biochim Biophys Acta 2009, 1787, 1433-1443, doi:10.1016/j.bbabio.2009.06.003.
4. Su, C.Y.; Huang, G.C.; Chang, Y.C.; Chen, Y.J.; Fang, H.W. Analyzing the Expression of Biomarkers in Prostate Cancer Cell Lines. In Vivo 2021, 35, 1545-1548, doi:10.21873/invivo.12408.
5. Molter, C.W.; Muszynski, E.F.; Tao, Y.; Trivedi, T.; Clouvel, A.; Ehrlicher, A.J. Prostate cancer cells of increasing metastatic potential exhibit diverse contractile forces, cell stiffness, and motility in a microenvironment stiffness-dependent manner. Front Cell Dev Biol 2022, 10, 932510, doi:10.3389/fcell.2022.932510.

Reviewer 2 Report (New Reviewer)
Comments and Suggestions for Authors
The authors have satisfactorily addressed my observations and have sincerely discussed the limitations of their work, complementing it with a strong defense of their results and additional research. I consider the article suitable for publication.
Author Response
1. The authors have satisfactorily addressed my observations and have sincerely discussed the limitations of their work, complementing it with a strong defense of their results and additional research. I consider the article suitable for publication.
Response: We sincerely appreciate your thorough evaluation of our manuscript and your positive feedback. We are grateful for your constructive comments, which have helped us strengthen our study through additional analyses and a clearer discussion of its limitations. Your insights have been invaluable in refining our work, and we are pleased that you consider the article suitable for publication. Thank you for your time, effort, and support throughout the review process.

Round 3
Reviewer 1 Report (Previous Reviewer 2)
Comments and Suggestions for Authors
I appreciate the author's efforts in incorporating new data by adding additional cell lines. Before acceptance, please enhance the quality of the Western blot images in Figure 5 and provide quantification from at least three independent experiments.
Author Response
Response to Reviewer 1:
We sincerely thank Reviewer 1 for their constructive feedback, which has helped us to improve the quality of our manuscript. Below, we address each comment point-by-point:
1. I appreciate the author's efforts in incorporating new data by adding additional cell lines. Before acceptance, please enhance the quality of the Western blot images in Figure 5 and provide quantification from at least three independent experiments.
Response: We sincerely appreciate the reviewer’s valuable feedback and their recognition of our efforts in incorporating additional cell lines. In response to the reviewer’s request, we have enhanced the quality of the Western blot images in Figure 5. Additionally, we have performed densitometric analysis from at least three independent experiments and included the quantification data, expressed as fold changes relative to the control, in the figure legends and bar graphs in Figures 5B and 5D. These revisions strengthen the reliability of our findings and address the reviewer’s concerns. We appreciate the constructive suggestions and believe that the improvements meet the journal’s standards for acceptance.

This manuscript is a resubmission of an earlier submission. The following is a list of the peer review reports and author responses from that submission.
Round 1
Reviewer 1 Report
Comments and Suggestions for Authors
In the present study, the authors examine the effect of Antrodia cinnamomea formula (XZF) on the viability of prostate cancer cells (PC3 cell line), macrophages, and T lymphocytes, as well as on the expression of PD-1 and PD-L1. While the topic of the study is interesting, I believe that the experiments conducted are insufficient to draw valuable conclusions. The authors only investigated the effect of different doses of XZF (50, 100, and 2000 µg/mL) on the aforementioned parameters (viability and expression of PD-1 and PD-L1). However, except for the use of a conditioned medium, the authors did not study the interaction between T lymphocytes, macrophages, and tumor cells. I believe that the authors should conduct functional assays to study the effect of XZF on different aspects of tumor cells (migration, invasion, etc.) to add value to the work. Additionally, the presentation of the results is not good, the figure legends are very incomplete, and the interpretation of the results is poor. Overall, I believe that the manuscript is not suitable for publication in IJMS.
Specific comments:
1. The title must be changed. The results do not support what the title suggests.
2. The 293T cell line is not mentioned in the methodology. Why was this cell line chosen and not another?
3. In the Legend of Figure 4, it is written "Figure 4. This is a figure. Schemes follow another format. If there are multiple panels, they should be listed as: (a) Description of what is contained in the first panel; (b) Description of what is contained in the second panel. Figures should be placed in the main text near to the first time they are cited."
Reviewer 2 Report
Comments and Suggestions for Authors
The manuscript submitted by Ming-Yen Tsai and colleagues investigates the immunomodulatory effects of Antrodia cinnamomea formula on a prostate cancer cell line. While the study explores an interesting topic, the experimental design requires significant improvement, and additional experiments are needed to substantiate the findings prior to publication. Currently, most conclusions are derived primarily from cell proliferation assays, which provide limited insight into the proposed immunomodulatory mechanism.
The primary issue pertains to the study's scope. If the aim is to determine whether Antrodia cinnamomea formula inhibits prostate cancer via immune modulation, utilizing only the androgen-independent prostate cancer cell line PC-3 is insufficient. Additional cell lines, and ideally in vivo confirmation, are highly recommended. Moreover, only one experiment was conducted using conditioned media from RAW 264.7 cells on PC-3 cells. Since RAW 264.7 cells are murine, using human macrophages would be more appropriate. The proliferation impact from this conditioned media was also limited.
Additionally, the study reports that the formula affects both CD4+ and CD8+ T-cell proliferation. Notably, an increase in CD8+ cytotoxic T-cell proliferation could potentially promote cancer progression. However, it is unclear why these experiments were conducted, as prostate cancer is a “cold” tumor and no subsequent analyses of T-cell activity in prostate cancer were included.
The use of 293T cells as non-tumorigenic controls is also questionable, as 293T cells are transformed and may not accurately represent non-cancerous cells.
Some formal aspects also require attention. For example, the last author appears to be missing in the author list, Figure 1 is not included, and the quality of the figures needs improvement.
Comments on the Quality of English LanguageThe manuscript's style should be enhanced to increase its appeal and readability for the audience.